# Bone mineral loss damages renal tubules in mice

Hirosaka Hayashi[1,2], Yutaka Miura[1,3], Yoshitaka Iwazu[3,4], Hideyuki Mukai[1,3], Yoshiyuki Mori[2], Takahiro Kuchimaru [5], Nobuhiko Ohno[6,7], Tatsuya Aiba [8], Risa Okada[9], Daisuke Kamimura[9], Dai Shiba [7], Hiroshi Kurosu[1] & Makoto Kuro-o [3] ✉

Fibroblast growth factor-23 (FGF23) is a bone-derived hormone that promotes urinary phosphate excretion in response to phosphate loading. While essential for phosphate homeostasis, elevated FGF23 increases phosphate concentration in the renal tubular fluid, promoting calcium-phosphate crystal formation and tubular injury. Here we show that bone resorption mobilizes phosphate into the circulation and mimics the pathophysiology of dietary phosphate loading. Enhanced bone resorption, induced by soluble receptor activator of NF-κB ligand (sRANKL) administration or microgravity exposure on the International Space Station, increased circulating FGF23 levels and caused renal tubular injury in mice. Pre-treatment with bisphosphonate, an inducer of osteoclast apoptosis, prevented sRANKL-induced increases in FGF23 and tubular damage. These findings suggest that bone mineral loss may contribute to renal tubular injury in clinical settings, including immobilization, osteoporosis, and chronic kidney disease–mineral bone disorder.

Oral ingestion of excess phosphate has been reported to cause kidney damage in rodents and humans[1–4]. When mice were fed a high phosphate diet containing 2.0% inorganic phosphate, they developed renal tubular damage and interstitial fibrosis within 4 weeks[1,2]. It has also been reported that phosphate-induced kidney damage correlates with the phosphate load excreted per nephron[1,5], indicating that mice/rats with lower nephron numbers develop more severe kidney damage when placed on the same high phosphate diet. However, the molecular mechanism behind these observations was not clear until recently.

We have shown that the kidney damage induced by dietary phosphate loading is associated with an increase in circulating levels of fibroblast growth factor-23 (FGF23)[1]. FGF23 is a phosphaturic hormone secreted by osteocytes/osteoblasts in response to dietary phosphate intake[6,7]. Phosphate adsorbed in the gastrointestinal tract flows into the blood and raises the blood phosphate levels, triggering the precipitation of amorphous (noncrystalline) calcium phosphate, which is adsorbed by the serum protein fetuin-A. This results in formation of mineral-protein complexes consisting of amorphous calcium-phosphate and a single molecule of fetuin-A, called calciprotein monomers (CPM). CPM have a particle size of 9 nm and

agglomerate to form colloids with particle sizes of tens of nm. These are called primary calciprotein particles (CPP1). Over time, the calcium-phosphate in CPP1 undergoes a phase transition from amorphous to crystalline and the agglomeration progresses to a particle size of several hundred nm. These large CPP containing crystalline calcium-phosphate are called secondary CPP (CPP2)[7–10]. The maturation process from CPM to CPP1 and then to CPP2 proceeds spontaneously over time and is accelerated at high phosphate concentration, calcium concentration, pH, and temperature, and decelerated at high concentrations of fetuin-A and magnesium. CPM and CPP1 induce FGF23 production/secretion in osteoblasts[11]. CPP2 have pathogenic activity that induces cell damage, innate immune responses, and vascular calcification[1,12–14].

The kidney is a major target organ of FGF23. The receptor for FGF23 is the fibroblast growth factor receptor (FGFR) complexed with Klotho protein, a single-pass transmembrane protein expressed on renal tubular cells[15–18]. Binding of FGF23 to the FGFR-Klotho complex induces phosphaturia by suppressing phosphate reabsorption at the proximal tubule[6,7]. Thus, FGF23 increases phosphate excretion per nephron and phosphate concentration in the proximal tubular fluid. When the phosphate

[1]Division of Anti-aging Medicine, Center for Molecular Medicine, Jichi Medical University, Tochigi, Japan. [2]Department of Dentistry, Oral and Maxillofacial Surgery, Jichi Medical University, Tochigi, Japan. [3]Division of Mineral Metabolism, Center for Molecular Medicine, Jichi Medical University, Tochigi, Japan. [4]Department of Clinical Laboratory Medicine, Jichi Medical University, Tochigi, Japan. [5]Division of Bioconvergence, Center for Molecular Medicine, Jichi Medical University, Tochigi, Japan. [6]Division of Histology and Cell Biology, Department of Anatomy, Jichi Medical University, Tochigi, Japan. [7]Division of Ultrastructural Research, National Institute for Physiological Sciences, Aichi, Japan. [8]Human Spaceflight Technology Directorate, Japan Aerospace Exploration Agency (JAXA), Ibaraki, Japan. [9]Space Environment Utilization Center, Human Spaceflight Technology Directorate, Japan Aerospace Exploration Agency (JAXA), Ibaraki, Japan. ✉e-mail: mkuroo@jichi.ac.jp

concentration exceeds solubility and calcium-phosphate precipitates in the proximal tubular fluid containing much lower concentrations of fetuin-A than in the blood, CPP rapidly mature to CPP2. CPP2 are captured by Toll-like receptor-4 (TLR4) expressed on the apical membrane of proximal tubules, endocytosed, and delivered to lysosomes[1]. However, CPP2 overload in renal tubular cells results in lysosomal dysfunction, impaired endosomal trafficking, and inflammasome activation, leading to tubular cell damage followed by inflammatory cell infiltration and interstitial fibrosis[19].

We reasoned that the entry of phosphate not only from the gastrointestinal tract (exogenous phosphate) but also from the bone (endogenous phosphate) into the bloodstream could cause renal damage by the same molecular mechanism. Specifically, we hypothesized that bone mineral loss could increase circulating FGF23 levels, induce phosphaturia, and damage renal tubules. To test this hypothesis, we stimulated bone resorption in mice and asked whether the pathophysiology observed in mice fed a high phosphate diet could be recapitulated.

## Results

### Bone resorption induces phosphaturia

Receptor activator of NF-κB ligand (RANKL) is a membrane protein expressed on the surface of osteoblasts[20]. RANKL induces osteoclast differentiation by binding to RANK expressed on the surface of osteoclast precursors. The recombinant extracellular domain of RANKL protein (sRANKL) induces osteoclast differentiation as a soluble factor[21–23]. Mice injected with a single dose of sRANKL showed an increase in plasma levels of tartrate-resistant acid phosphatase type 5b (TRACP-5b) within 18 h, indicating increased osteoclast activity (Fig. 1a). Concurrently, the trabecular bone volume over the tissue volume (BV/TV) in the femurs showed a decreasing trend up to 24 h after the sRANKL injection, although this change did not reach statistical significance (Fig. 1b). As the bone mineral density was not changed (Fig. 1c), we hypothesized that a statistically significant loss of bone mineral could be achieved by intensifying the sRANKL treatment. Nonetheless, plasma levels of phosphate (Fig. 1d) and calcium (Supplementary Fig. S1A) showed a transient but statistically significant elevation. Correspondingly, calcium phosphate products were significantly increased (Supplementary Fig. S1B), which was associated with a transient increase in plasma CPP levels at 18 h (Fig. 1e). Plasma FGF23 levels first showed a statistically significant increase at 21 h and remained elevated until 30 h after the sRANKL injection (Fig. 1f). Subsequently, urinary phosphate excretion was increased (Fig. 1g, h). All of these changes returned to baseline within 48 h after sRANKL injection. The phosphaturia was caused by the increase in FGF23 but not parathyroid hormone (PTH), because plasma PTH levels were decreased after the sRANKL injection (Supplementary Fig. S1C). Consistent with the increased FGF23, renal expression of the *Cyp27b1* gene was decreased (Supplementary Fig. S1D), whereas expression of the *Cyp24a1* gene was reciprocally increased (Supplementary Fig. S1E)[6], which may explain a decreasing trend in plasma levels of active vitamin D (Supplementary Fig. S1F). Urinary calcium excretion was increased within 18 h after the sRANKL injection (Supplementary Fig. S1G, H). This may be explained by the fact that plasma levels of PTH, which inhibits urinary calcium excretion, reached its peak decrease within 18 h after sRANKL injection (Supplementary Fig. S1C). In conclusion, the sRANKL-induced increase in bone resorption mobilized phosphate and calcium from bones into the systemic circulation. The increase in blood calcium phosphate products triggered CPP formation, which in turn raised plasma FGF23 levels to induce phosphaturia, restoring the blood phosphate levels to the baseline.

### Sustained bone resorption induced renal tubular damage

Although the single dose of sRANKL induced phosphaturia, overt renal tubular damage was not detectable. Therefore, we treated mice with sRANKL every 24 h for three consecutive days (the triple dose regimen) to stimulate bone resorption for a longer period of time[23]. Mice treated with the triple dose regimen had higher plasma TRACP-5b levels (Fig. 2a) and less trabecular bone than those treated with the single dose regimen within 1.5 h after the last sRANKL injection (Fig. 2b), which indicated a significant bone

mineral loss because the bone mineral density remained constant as determined by μCT (Fig. 2c). Plasma levels of phosphate (Fig. 2d) and calcium (Supplementary Fig. S2A) began to increase within 1.5 h after the last sRANKL injection. Correspondingly, plasma calcium phosphate products (Supplementary Fig. S2C) and plasma CPP levels (Fig. 2e) were increased, followed by an increase in plasma FGF23 levels (Fig. 2f) that was faster and more robust than that seen with the single dose regimen. Metabolic cage analysis (Fig. 2g) showed a progressive increase in urinary phosphate excretion up to 24 h after the last sRANKL injection (day 3) (Fig. 2h). As in the single dose regimen, plasma PTH levels were significantly decreased (Supplementary Fig. S2C). The increase in FGF23 explains a decrease in *Cyp27b1* gene expression, an increase in *Cyp24a1* gene expression, and a decreasing trend in plasma active vitamin D levels (Supplementary Fig. 2D–F). The bone histomorphometric analysis demonstrated a significant increase in bone resorption and a compensatory increase in bone formation, resulting in a net bone loss (Supplementary Fig. S3). This rapid and robust increase in bone resorption by the triple dose regimen is likely because the preceding two doses of sRANKL had induced the commitment of osteoclast precursors to osteoclasts.

We measured the renal expression of tubular damage markers by quantitative RT-PCR and observed a significant increase in the mRNA levels of *osteopontin* (Fig. 3a), *neutrophil gelatinase-associated lipocalin* (*Ngal*, Fig. 3b), *kidney injury molecule-1* (*Kim-1*, Fig. 3c), and *interleukin-36α* (*IL-36α*, Fig. 3d). However, the renal tubular damage was not associated with an increase in plasma creatinine levels (Fig. 3f) or urinary glucose excretion (Supplementary Fig. S4), suggesting that the transient tubular injury induced by sRANKL treatment did not lead to significant impairment of renal function. Immunohistochemical analysis confirmed the expression of osteopontin and IL-36α protein in the renal tubules at the cortico-medullary junction with a segmented distribution along the radial tracts of the nephrons (Fig. 3g). This expression profile of osteopontin and IL-36α was identical with that seen in mice fed a high phosphate diet[1]. The expression of osteopontin and IL-36α at the mRNA and the protein levels peaked at 24 h and 36 h after the sRANKL injection, respectively. Consistent with the increased expression of tubular damage markers, histological analysis revealed tubulointerstitial damages characterized by vacuolar degeneration and interstitial cell infiltration in sRANKL-treated mice but not in the vehicle-treated mice (Supplementary Fig. S5A, B). In addition, ex vivo imaging detected CPP in the proximal tubular lumen in the sRANKL-treated mice but not in the vehicle-treated mice (Supplementary Fig. S5C, D).

### sRANKL treatment induced sustained kidney injury in mice fed a high-phosphate diet

As we previously reported, mice fed a high-phosphate diet over several weeks develop renal injury characterized by tubular damage and interstitial inflammation/fibrosis[1,2]. In contrast, mobilization of bone-derived endogenous phosphate by sRANKL treatment alone elicited only a transient upregulation of inflammatory and tubular injury markers (Fig. 3). We therefore hypothesized that, in mice subjected to dietary (exogenous) phosphate loading, additional endogenous phosphate loading from bone triggered by sRANKL treatment may act as a "second hit" that prolongs and exacerbates kidney injury. To test this, mice were fed either a high-phosphate diet (1.5% inorganic phosphate) or a normal-phosphate diet (0.35% inorganic phosphate) and received sRANKL or vehicle according to the triple-dose regimen administered every other week (Fig. 4a). As expected, sRANKL treatment alone did not maintain elevated tubular injury markers 10 days after the final injection. The high-phosphate diet alone significantly increased tubular injury markers (*osteopontin, IL-36α, Kim-1*, and *Ngal*), whereas inflammatory markers (*F4/80, Mcp-1, TNFα*, and *IL-6*) showed upward trends without reaching statistical significance (Fig. 4b–e; Supplementary Fig. S6A–D), and the fibrosis marker collagen 1a1 was not upregulated (Fig. 4f). In contrast, the combination of sRANKL treatment and high-phosphate diet feeding produced persistent upregulation of *osteopontin, IL-36α, F4/80, Mcp-1,* and *collagen 1a1* even 10 days after the

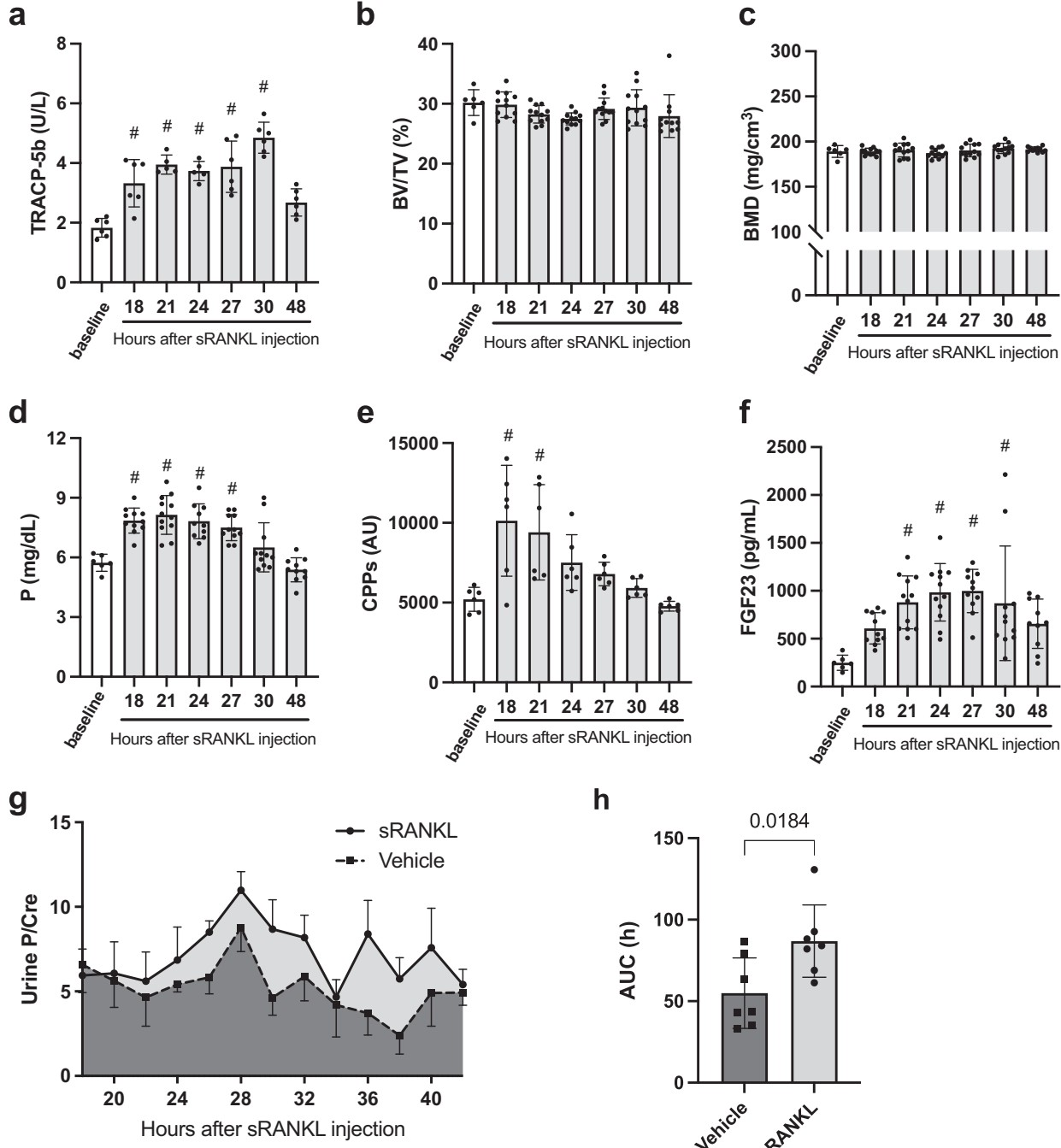

**Fig. 1 | Effects of the single dose sRANKL on the bone and phosphate metabolism.** Mice were sacrificed before (baseline) or after intraperitoneal administration with sRANKL (2 mg/kg) at the indicated time points to harvest femur and blood. **a** Plasma TRACP-5b levels. **b** Trabecular bone volume fraction (BV/TV) representing the volume of mineralized bone (BV) per unit volume of the total bone marrow space (TV). **c** Bone mineral density (BMD). Plasma levels of phosphate (**d**), CPPs (**e**), and FGF23 (**f**). Data are presented as means ± SD. $N$ = 5–6 for each column. $^\#P < 0.01$ versus baseline by one-way ANOVA with Tukey's multiple comparison test. **g** Urine phosphate/creatinine ratios (Urine P/Cre). Spot urine samples were collected after intraperitoneal administration of sRANKL (solid circles) or vehicle (solid squares) every 2 h at the indicated time points. Data are presented as means ± SEM. $N$ = 3–7 for each column. **h** Area under the curve (AUC) of the urine phosphate/creatinine ratios in individual mice injected with vehicle or sRANKL. Data are presented as means ± SD. $N$ = 7 for each column. $P = 0.0184$ versus vehicle by Student's $t$ test.

final sRANKL injection, with stronger effects than those induced by the high-phosphate diet alone (Fig. 4b–f). Circulating FGF23 levels were also further increased by sRANKL treatment (Fig. 4g). Two-way ANOVA revealed a significant main effect of high-phosphate diet and sRANKL treatment in addition to a substantial interaction between them as determined by partial $\eta^2$, indicating that sRANKL-mediated endogenous phosphate mobilization can induce sustained and additional FGF23 increase, tubular injury, and interstitial inflammation/fibrosis in the context of

dietary (exogenous) phosphate loading. Consistently, histological analyses demonstrated pronounced kidney injury, including tubular vacuolization, inflammatory cell infiltration, interstitial fibrosis, and increased expression of tubular damage markers (Supplementary Fig. S7).

## Effects of microgravity on the bone and kidney

In the microgravity environment of spaceflight, bone loss occurs in both astronauts and mice[24]. We hypothesized that microgravity induced bone

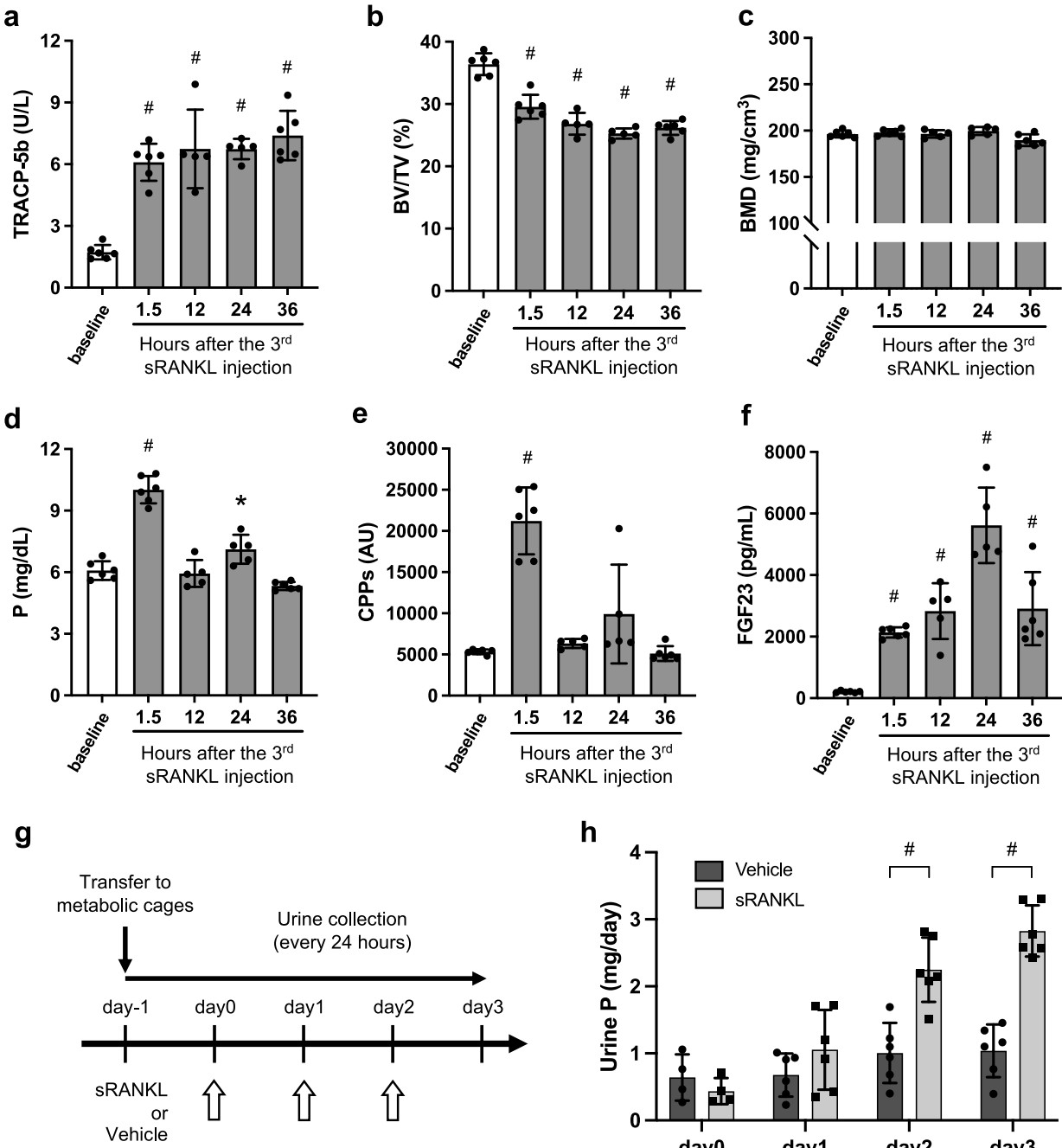

**Fig. 2 | Effects of the triple dose sRANKL on the bone and phosphate metabolism.** Mice were sacrificed before (baseline) or after intraperitoneal administration with sRANKL (2 mg/kg, every 24 h for 3 times) at the indicated time points to harvest kidney, femur, and blood. The kidneys were used in Fig. 3 and Supplementary Fig. 2. **a** Plasma TRACP-5b levels. **b** Trabecular bone volume fraction (BV/TV). **c** Bone mineral density (BMD). Plasma levels of phosphate (**d**), CPPs (**e**), and FGF23 (**f**). Data are presented as means ± SD. $N = 5$–6 for each column. $^{*}P < 0.05$, $^{\#}P < 0.01$ versus baseline by one-way ANOVA with Tukey's multiple comparison test. **g** A scheme of the metabolic cage study. **h** The amount of urinary phosphate excretion per day (Urine P) in mice administered with vehicle (black columns, solid circles) or sRANKL (gray columns, solid squares). Data are presented as means ± SD. $N = 6$ for each column. $^{\#}P < 0.01$ versus vehicle by one-way ANOVA with Tukey's multiple comparison test.

loss would cause a pathophysiology similar to that induced by sRANKL administration. To test this hypothesis, we sent mice to the ISS to expose them to the microgravity environment for 9–10 days. As observed in the sRANKL-treated mice, the mice that stayed on the ISS (the microgravity group) had higher plasma levels of phosphate, calcium, calcium-phosphate product, CPP, and FGF23 than the mice housed on ground (the normal gravity group) (Fig. 5a–e). As the food intake was matched between the groups, these observations suggest that exposure to microgravity increased the influx of phosphate and calcium into the blood not from the exogenous

sources (food) but from endogenous sources (tissues), most likely from bone. Indeed, the microgravity group had significantly higher plasma TRACP-5b levels than the normal gravity group, indicating an increase in bone resorption. Although the μCT analysis and bone histomorphometric analysis failed to detect statistically significant difference in bone morphology between the two groups, bone transcriptome analysis supported a decrease in bone formation and an increase in bone resorption in the microgravity group (Supplementary Data 2). Specifically, genes upregulated by microgravity were enriched in the reactomes relevant to bone loss. The

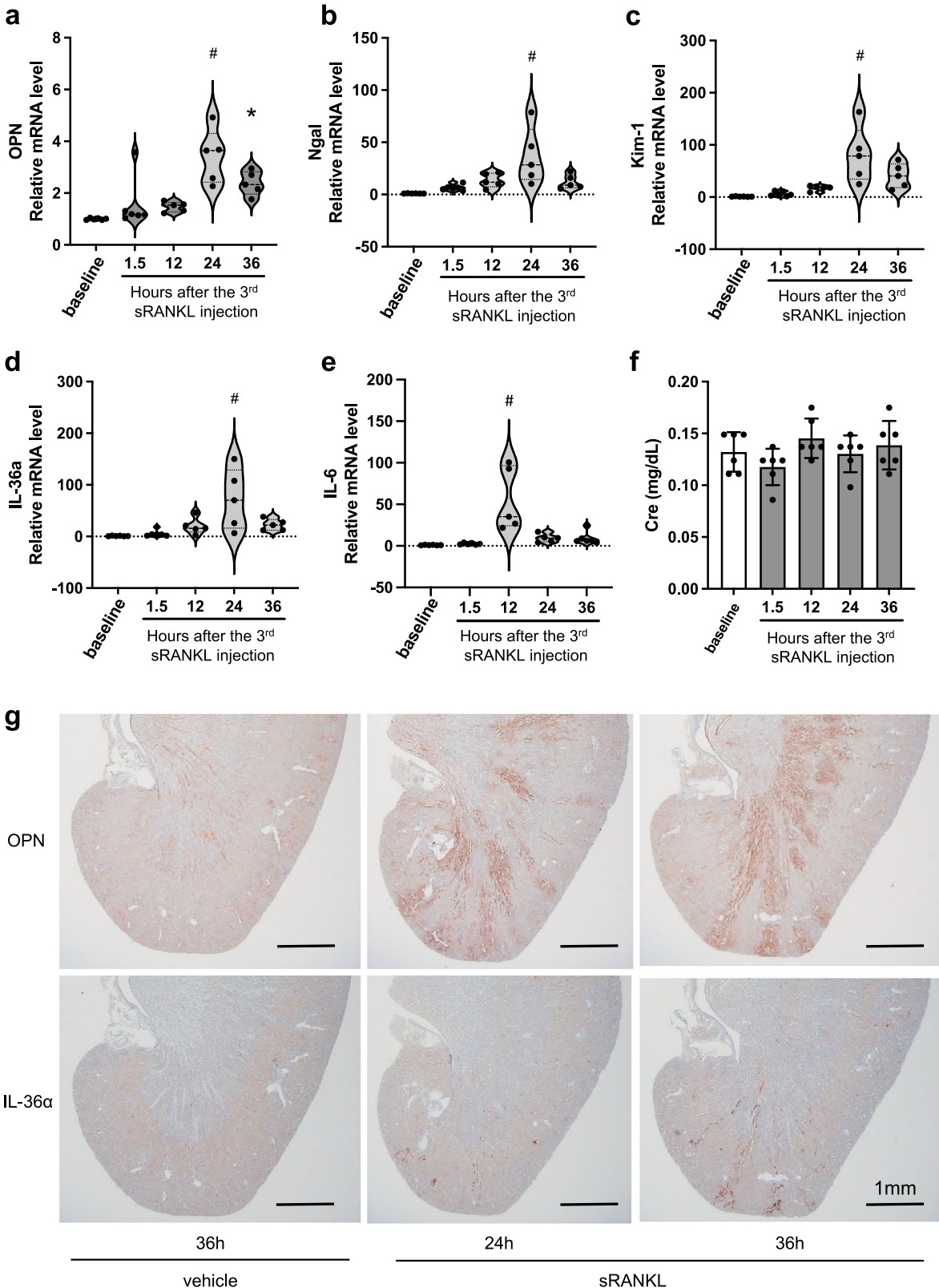

**Fig. 3 | Effects of the triple dose sRANKL on the kidney.** Total RNA and formalin-fixed paraffin sections were prepared from the kidneys from mice used in Fig. 2. Relative mRNA levels of renal tubular damage markers were measured by quantitative RT-PCR. **a** Osteopontin. **b** Neutrophil gelatinase-associated lipocalin (Ngal). **c** Kidney injury molecule-1 (Kim-1). **d** Interleukin-36α (IL-36α). **e** Interleukin-6 (IL-6). Data are indicated as violin plots with the median and quartiles (dotted lines). $N = 6$ for each column. $*P < 0.05$, $^{\#}P < 0.01$ versus baseline by one-way ANOVA with Tukey's multiple comparison test. **f** Plasma creatinine levels. Data are presented as means ± SD. $N = 6$ for each column. No statistical differences were observed between the groups by one-way ANOVA with Tukey's multiple comparison test. **g** Representative kidney section images of immunohistochemistry using antibodies against osteopontin (upper panels) or IL-36α (lower panels) at the indicated time points after the last sRANKL or vehicle injection. Bar = 1 mm.

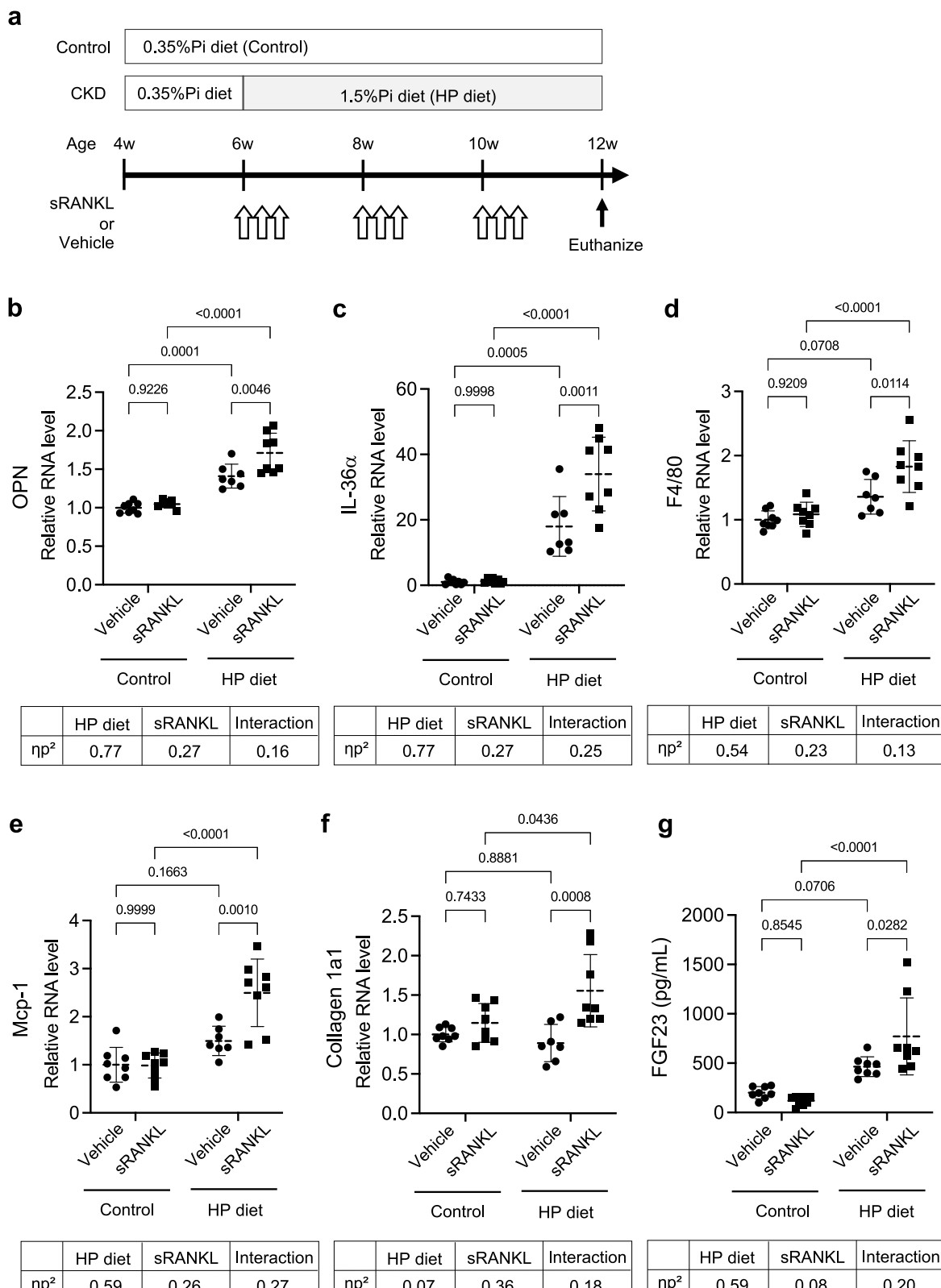

**Fig. 4 | Effects of sRANKL treatment on kidney injury in mice fed a high-phosphate diet. a** The study design. Exogenous phosphate loading was induced by feeding a high phosphate diet (1.5% inorganic phosphate, "Pi", HP diet) for 6 weeks, and endogenous phosphate loading was elicited by sRANKL administration using the triple dose regimen given every other week, respectively. **b–f** Relative mRNA levels of *osteopontin* (*OPN*), *IL-36α, F4/80, Mcp-1*, and *Collagen 1a1* quantified by qPCR. **g** Plasma FGF23 levels. Data are presented as means ± SD. $N = 7–8$ per group. Statistical significance was assessed by two-way ANOVA followed by Tukey's multiple-comparison test; *P* values are indicated. Effect sizes of HP diet, sRANKL treatment, and their interaction were calculated from the ANOVA sums of squares and shown as partial $\eta^2$ ($p\eta^2$).

**Fig. 5 | Effects of microgravity on the kidney and phosphate metabolism.** Plasma levels of phosphate (**a**), calcium (**b**), calcium phosphate product (**c**), CPPs (**d**), FGF23 (**e**), and TRAcP-5b (**f**) of the mice from the normal gravity group (1 G) and the microgravity group (0 G). Relative mRNA levels of osteopontin (**g**) and interleukin-36α (IL-36α) (**h**) in the kidney. Data are presented as means ± SD. $N = 6$ for each column. $P$ values by $t$ test are indicated.

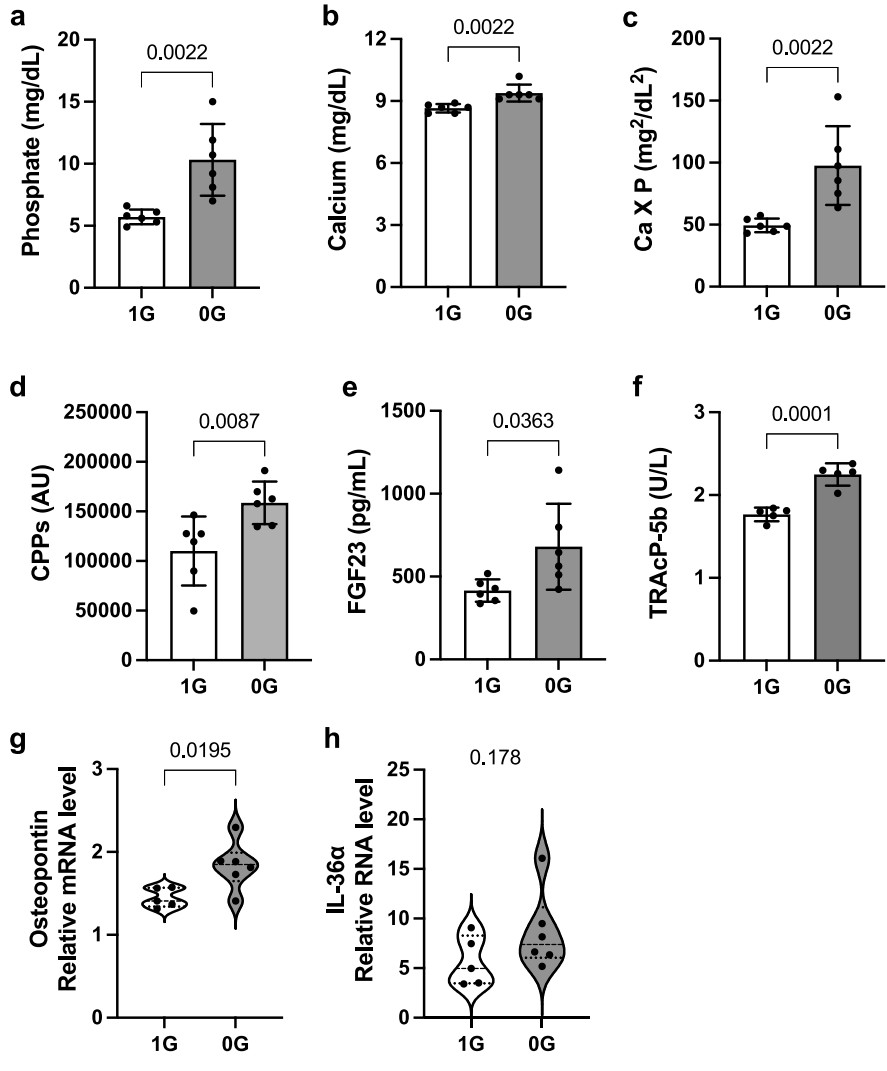

most prominent enrichment was the PRC2 (polycomb repressive complex 2)-mediated epigenetic modification, which is known to inhibit osteogenic differentiation and activate osteoclasts[25]. It also promotes adipogenic differentiation, potentially inhibiting osteogenic differentiation by facilitating commitment of mesenchymal stem cells to the adipogenic lineage rather than the osteogenic lineage[26]. In addition, interleukin-7 signaling was significantly enriched, which is known to inhibit osteogenic differentiation and reciprocally promote osteoclast differentiation[27,28]. Metalloprotease DUBs (deubiquitinases) were also enriched, which contribute to bone remodeling[29] (Supplementary Fig. S8A). Enriched transcription factors included CCAAT enhancer binding protein β (*Cebpb*), nuclear factor κB subunit 1 (*Nfkb1*), and peroxisome proliferator-activated receptor γ (*Pparg*), each of which is known to suppress osteoblast differentiation, stimulate osteoclast differentiation, and promote adipogenic differentiation[30–32] (Supplementary Fig. S8B). Conversely, genes downregulated by microgravity were particularly enriched in the assembly of collagen fibrils required to maintain the bone matrix structure necessary for bone mineralization[33] (Supplementary Fig. S9A). Other enriched transcription factors included Runt-related transcription factor 2 (*Runx2*) and its downstream target *Sp7/Osterix*, which are essential for osteoblast differentiation[34] (Supplementary Fig. S9B). Taken together, the transcriptome analysis supports that the exposure to microgravity for 9–10 days accelerated bone resorption and suppressed bone formation, leading to a net bone loss. These changes in the gene expression profile were associated with renal tubular damage as determined by a significant increase in osteopontin expression (Fig. 5g). An

increasing trend in expression levels of another tubular damage marker interleukin-36α was observed, although this change did not reach statistical significance (Fig. 5h).

**Suppression of bone resorption prevented renal tubular damage**
To confirm that the renal tubular damage was dependent on phosphate released from the bone, we suppressed sRANKL-induced bone resorption by risedronate, a bisphosphonate that induces osteoclast apoptosis (Fig. 6a). The pretreatment with risedronate significantly attenuated the increase in osteoclast activity (Fig. 6b) and prevented the bone mineral loss induced by sRANKL (Fig. 6c, d). We confirmed that the risedronate treatment did not down-regulate RANK expression (Fig. 6e), indicating that risedronate blocked the sRANKL-induced bone mineral loss not through blocking RANK expression in osteoclast precursors. Accordingly, the mice co-treated with sRANKL and risedronate had lower plasma FGF23 levels (Fig. 6f) and less urinary phosphate excretion (Fig. 6g) than the mice treated with sRANKL alone. These changes induced by the risedronate treatment were associated with significant decrease in expression of markers for renal tubular damage and inflammation at the mRNA and protein levels (Fig. 7). Consistent with this, the risedronate treatment alleviated sRANKL-induced interstitial cell infiltration (Fig. 7f), whereas it did not affect either plasma levels of phosphate, calcium, and creatinine or renal expression levels of the *Cyp24a1* and *Cyp27b1* genes in mice treated with sRANKL (Supplementary Fig. S10).

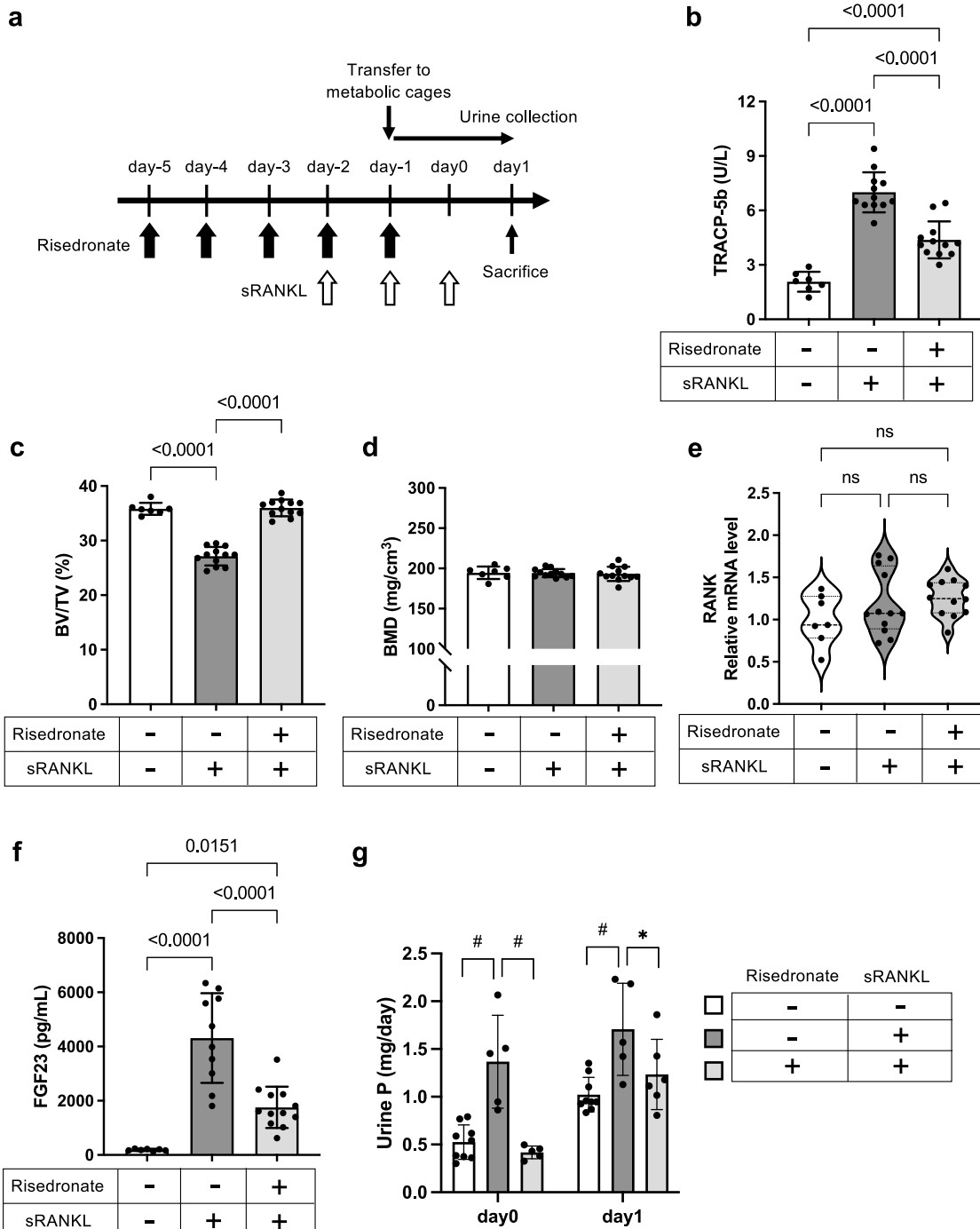

**Fig. 6 | Risedronate counteracts the effects of the triple dose sRANKL on the bone and phosphate metabolism. a** The study design. Mice were subcutaneously injected with risedronate (10 μg/kg) or vehicle every 24 h on day-5 and thereafter through day-1. The triple dose of sRANKL or vehicle was started on day-2. All the mice were transferred individually to metabolic cages on day-1 to collect urine for two days and then sacrificed on day 1 to harvest kidney, femur, and blood. The kidneys were used in Fig. 7. Three groups (vehicle treatment alone, sRANKL treatment alone, and co-treatment with risedronate and sRANKL) were set up. **b** Plasma TRACP-5b levels. **c** Trabecular bone volume fraction (BV/TV). **d** Bone mineral density (BMD). Data are presented as means ± SD. $N = 6$ for each column. $P$ values by one-way ANOVA with Tukey's multiple comparison test are indicated when significant. **e** Relative

mRNA levels of RANK in the femurs determined by quantitative RT-PCR. The residronate treatment did not affect expression levels of RANK. Data are indicated as violin plots with the median and quartiles (dotted lines). $N = 7$–12 for each column. No statistical differences were observed by Kruskal–Wallis's test with Dunn's multiple-comparison test. **f** Plasma FGF23 levels. Data are presented as means ± SD. $N = 7$–12 for each column. $P$ values by one-way ANOVA with Tukey's multiple comparison test are indicated. **g** The amount of urinary phosphate excretion per day (Urine P) in three different treatment groups. Data are presented as means ± SD. $N = 5$–9 for each column. $^{\#}P < 0.01$ by one-way ANOVA with Tukey's multiple comparison test.

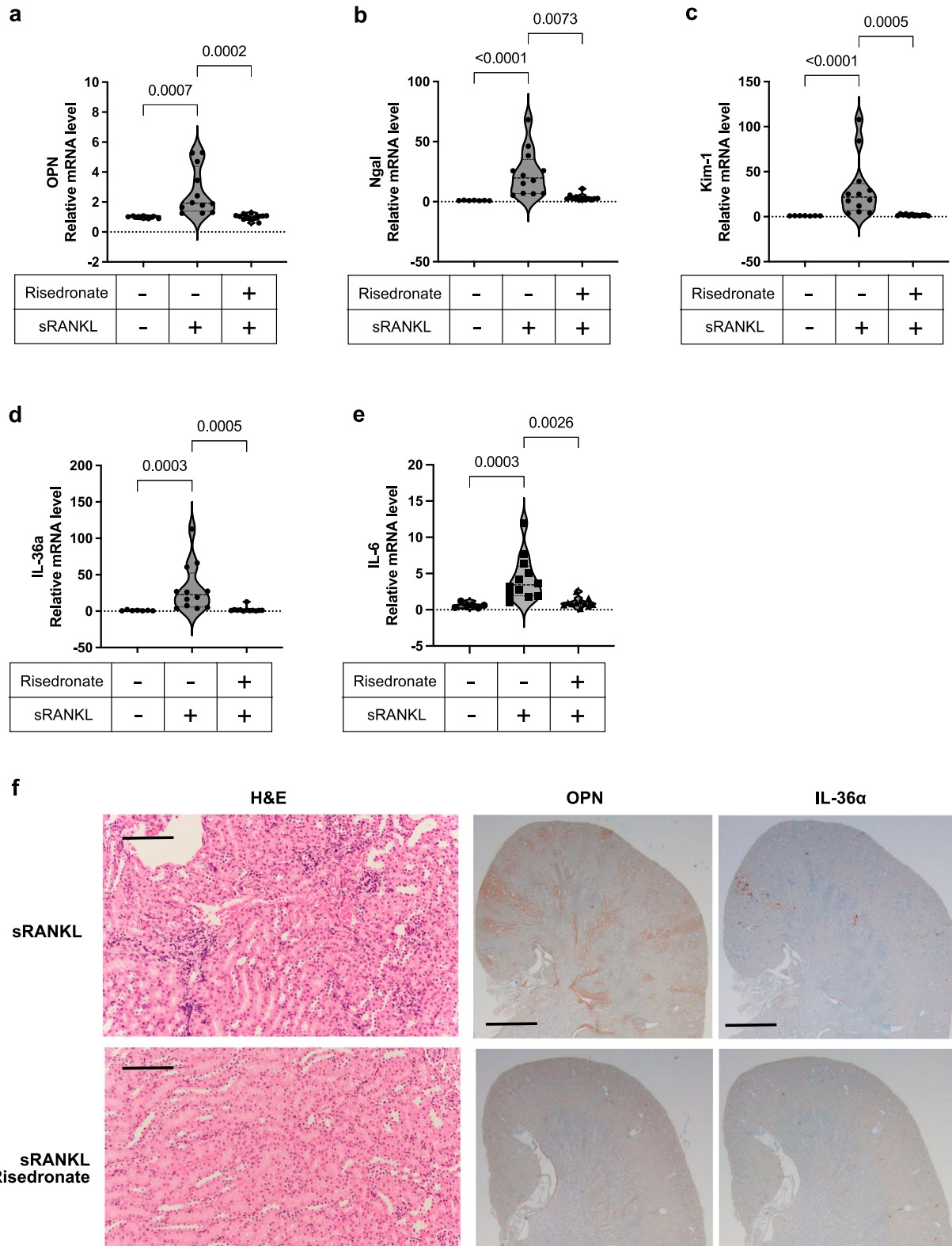

**Fig. 7 | Risedronate counteracts the effects of the triple dose sRANKL on the kidney.** Total RNA and formalin-fixed paraffin sections were prepared from the kidney samples from mice in Fig. 6. Relative mRNA levels of *Osteopontin* (**a**), *Ngal*, (**b**), *Kim-1* (**c**), *IL-36α* (**d**), and *matrix metalloprotease-3* (*MMP3*) (**e**) were measured by quantitative RT-PCR. Data are indicated as violin plots with the median and quartiles (dotted lines). *N* = 7–12 for each column. *P* values by Kruskal–Wallis's test

with Dunn's multiple-comparison test are indicated. **f** Representative kidney section images of hematoxylin-eosin staining (H&E) and immunohistochemistry using antibodies against osteopontin (OPN) or IL-36α from mice treated with sRANKL (upper panels) and mice co-treated with sRANKL and Risedronate (lower panels). Bar: 0.1 mm for H&E and 1 mm for immunohistochemistry.

## Discussion

The present study has shown that an acute bone resorption can cause renal tubular damage in mice without significant increase in blood creatinine levels. This condition may be equivalent to subclinical acute kidney injury (AKI) in clinical settings[35]. Rapid bone mineral loss is typically observed in patients under bed rest. Immobilization during bed rest was reported to reduce bone mineral within a week because of greater bone resorption than bone formation, which preferentially affected trabecular bone[36]. AKI is highly prevalent among hospitalized patients under intensive care environment after cardiovascular surgery and associated with high morbidity[37]. Other clinical conditions causing bone mineral loss due to local and/or systemic immobilization (disuse osteoporosis) include cerebrovascular disease and fracture[38]. Indeed, AKI is one of the common complications in patients after hip fracture surgery[39]. Although primary causes of AKI may vary, bone mineral loss due to immobilization may be a secondary but universal cause of renal tubular damage potentially contributing to poor renal outcome in AKI patients. Of note, astronauts during spaceflight were reported to have increased bone resorption and unchanged/decreased bone formation, yielding a net bone mineral loss[40]. Furthermore, astronauts are known to have high risk for renal stone formation[41], which can be reduced by use of bisphosphonate[42]. It is plausible that calcium-phosphate microcrystals precipitated in the renal tubular fluid upon bone mineral loss may serve as nuclei for renal stones. It remains to be determined whether astronauts may develop subclinical AKI under microgravity. It also remains to be determined whether chronic bone mineral loss in patients with postmenopausal or senile osteoporosis may cause renal tubular damage.

Although bone mineral loss induced by sRANKL treatment caused only a transient increase in tubular damage markers in normal mice, it provoked persistent and exacerbated renal injury when combined with dietary phosphate loading. These findings are consistent with our previous study demonstrating that renal injury correlates with the phosphate load excreted per nephron[1]. In normal mice with preserved nephron number, phosphate released from bone by sRANKL caused only transient tubular damage. However, in mice already burdened by dietary (exogenous) phosphate loading and reduced nephron number, the same bone-derived endogenous phosphate acted as a "second hit," driving sustained tubular injury, inflammation, and fibrosis. These results underscore a clinically relevant mechanism in which renal vulnerability depends not only on dietary phosphate intake but also on bone-derived phosphate release and residual nephron reserve. This concept has important implications for patients with chronic kidney disease (CKD), who often experience enhanced bone resorption due to osteoporosis, hyperparathyroidism, immobility, or other high-turnover bone states. In such contexts, bone mineral loss may synergize with dietary phosphate load and diminished nephron number to accelerate CKD progression.

Bone is the major reservoir of phosphate that contains ~80% of the total body phosphate, while phosphate in the blood represents less than 0.3%[43]. Therefore, even a modest decrease in bone mineral can cause a significant increase in the blood phosphate level. Indeed, blood phosphate levels were significantly increased in the mice exposed to the microgravity only for 9–10 days (Fig. 4a) despite the fact that the morphological analyses did not detect any significant bone mineral loss. We assume that bone mineral loss and dietary phosphate load share the same mechanism of renal tubular damage, because both increase phosphate excretion per nephron. Namely, sustained entry of either exogenous phosphate from diet or endogenous phosphate from bone into the bloodstream increases plasma CPP and FGF23 levels to induce phosphaturia. This series of event is regarded as an adaptive response required for maintaining the blood phosphate level constant. However, FGF23-induced phosphaturia increases phosphate excretion per nephron to induce tubular damage through triggering formation of calcium-phosphate microcrystals in the tubular fluid[1].

After promotion of bone resorption by sRANKL administration, the rise in plasma CPP levels preceded the significant increase in FGF23 and that the FGF23 levels remained elevated even after the CPP levels had declined (Fig. 1e, f). This temporal association was consistent with the finding that CPP induced FGF23 expression, which we have reported previously[11] and therefore did not include the original experimental data supporting this finding in the present study. The decrease in PTH after the sRANKL injection (Supplementary Figs. S1C, S2C) may be due not only to the increase in plasma calcium levels (Supplementary Figs. S1A, S2A) but also to the increase in plasma FGF23 levels (Figs. 1f and 2f) as FGF23 suppresses PTH expression and secretion[44].

Although increased blood influx of phosphate from either an exogenous (dietary) or endogenous (bone) source can increase plasma FGF23 levels, the time course of FGF23 increase was significantly different between them. Bone resorption induced by a single dose of sRANKL raised plasma FGF23 levels robustly from ~200 pg/mL to ~1000 pg/mL and quickly within 18–24 h (Fig. 1). In contrast, a bolus phosphate administration by oral gavage in mice raised plasma FGF23 levels from ~200 pg/mL to 300 pg/mL at most[11]. Furthermore, it took a few days before dietary phosphate load raised plasma FGF23 levels to ~1000 pg/mL[1,11]. The difference in the time course can be explained by the fact that CPM and CPP1, but not phosphate per se, stimulate FGF23 production in osteocytes/osteoblasts. CPPs can extravasate through sinusoids in the bone marrow[11]. However, CPM with a particle size of 9 nm have another clearance pathway; CPM can be filtrated through the glomeruli and rapidly removed from the blood. Therefore, CPM may not reach osteocytes/osteoblasts in vivo. However, if CPM formation is accelerated by dietary phosphate loading and/or if the CPM clearance is delayed by impaired renal function, blood CPM levels will be elevated and the likelihood of CPM extravasation through the bone marrow sinusoids may be increased. There may also be an increased likelihood of CPP1 formation in the blood. CPP1 cannot pass through the glomeruli due to its large particle size but can extravasate through the sinusoids. CPM and CPP1 extravasating from the bone marrow sinusoids can act on osteocytes/osteoblasts to stimulate FGF23 production. Therefore, in order to increase blood FGF23 levels with dietary phosphate loading, it is necessary to increase the formation of CPM to an extent that exceeds the renal clearance of CPM by sustained dietary phosphate loading. In contrast, the promotion of osteoclast differentiation and bone resorption by sRANKL administration can increase local concentrations of calcium and phosphate in the interstitial fluid of the bone marrow, facilitating the formation of CPM and CPP1 at the site, which act directly on osteocytes/osteoblasts to induce a rapid and robust increase in FGF23.

It was reported that exposure of mice to microgravity for 15 days induced about 6% reduction of BV/TV by μCT analysis[45]. In the present study, however, μCT analysis and bone morphometric analysis of the femurs and vertebrae failed to detect a significant difference in BV/TV between the microgravity and the normal gravity groups. In the present study, the bone loss caused by the short 9-10 day stay on the ISS was detectable by blood biochemistry and gene expression profiling, but may not have been robust enough to be detected morphologically.

Besides CPP, glycerol-3-phosphate (G3P) has been identified as an inducer of FGF23 production[46] To assess whether circulating G3P are increased and thereby contribute to the elevation of blood FGF23 in spaceflight mice, we performed plasma metabolome analysis (Supplementary Data 3). Plasma G3P levels were significantly increased in spaceflight mice (Supplementary Fig. S11A). However, hemolysis was present in 3 of 6 spaceflight plasma samples. Hemolysis releases glycerophosphorylcholine (GPC) from erythrocytes and potentially promotes enzymatic hydrolysis of GPC to G3P and choline[47,48]. Consistent with this, the hemolyzed samples showed higher G3P, choline, and GPC than non-hemolyzed samples (Supplementary Fig. S11B). Together, these observations suggest that the apparent increase in G3P in spaceflight mice may, at least in part, reflect a hemolysis-related artifact in the present study. It remains to be determined whether G3P contributes to an increase in circulating FGF23 levels during space flight. Impaired renal function is also known to be associated with elevated blood FGF23 levels[49]. Plasma metabolome analysis showed no differences in creatinine or urea levels between spaceflight and ground controls (Supplementary Fig. S11A), indicating that, within the sensitivity of these markers, renal function was not detectably impaired by the spaceflight.

Spaceflight and ground conditions differ not only in gravity but also in factors such as cosmic radiation, body fluid distribution, energy expenditure, and gastrointestinal physiology. These factors could, in principle, influence mineral metabolism and nutrient absorption despite matched food intake. Despite these limitations, we consider that the data obtained in the present space experiment are consistent with our hypothesis that phosphate released from bone contributes to the induction of renal tubular injury. The spaceflight mice exhibited significantly higher circulating phosphate, calcium, CPP, and FGF23 levels than the ground controls, indicating greater net mineral entry into the circulation. In addition, spaceflight mice showed higher plasma TRAcP-5b and a transcriptomic signature consistent with increased osteoclast activity and reduced bone formation, indicating increased release of endogenous phosphate and calcium from bones. Furthermore, prior study showed that simulated galactic cosmic rays alone promoted osteoclastogenesis and bone loss under normal gravity[50]. Thus, under space conditions, both microgravity and cosmic radiation contribute to enhanced bone resorption.

Despite this, one could argue that the intestinal absorption of nutrients in general, or minerals specifically, may be significantly increased in spaceflight mice for reasons that are not yet fully understood. This enhanced absorption contributes more to the increased mineral influx into the circulation than does enhanced bone resorption. However, metabolome analysis revealed that the spaceflight mice exhibited lower plasma levels of citrulline—a recognized biomarker of intestinal enterocyte mass and absorptive function[51]—indicating a likely reduction, rather than enhancement, of nutrient absorption capacity (Supplementary Fig. S11A). Moreover, microbiota–host co-metabolites such as p-hydroxybenzoate and phenaceturate were also significantly decreased (Supplementary Fig. S11A), a pattern consistent with diminished intestinal uptake. Collectively, these findings do not support a hypothesis that spaceflight mice experience a generalized enhancement of intestinal nutrient absorption. Regarding mineral absorption, calcium isotope and tracer studies in astronauts have shown that spaceflight reduces intestinal calcium absorption[52]. If the same holds true for mice, the observation that spaceflight mice nevertheless exhibited higher blood calcium levels suggests that bone-derived calcium influx exceeded any reduction in intestinal uptake. Because bone resorption releases both calcium and phosphate into the circulation, it is reasonable to infer that a substantial amount of phosphate was likewise mobilized from bone in spaceflight mice.

Taken together, under conditions of matched dietary intake, our findings support the interpretation that elevated blood phosphate and calcium in spaceflight mice are attributable primarily to increased bone resorption rather than to enhanced intestinal absorption. However, we did not directly measure intestinal phosphate absorption due to the logistical constraints of spaceflight experiments. In particular, urine collection is infeasible under microgravity. We have acknowledged this as a study limitation and propose that future investigations employ isotope tracers and fecal balance analyses to directly assess phosphate absorption under spaceflight versus ground conditions.

A new experimental protocol in the present space experiment was that the mice were sacrificed and frozen on the ISS. In all the previous space experiments carried out so far, the mice were brought back to Earth alive from the ISS. However, it usually took a few days to transport the mice to the laboratory for dissection. A major concern of this protocol is that the changes that had occurred in the microgravity environment could be reversed by the few days of exposure to normal gravity after returning to Earth. To eliminate this possibility, we euthanized the mice on the ISS, immediately froze the carcasses and the blood samples in the on-board freezer, and returned them frozen to Earth. Indeed, we have identified several microgravity-associated enrichment of reactomes not reported in mouse bone, including the PRC2-mediated methylation of histones/DNA and IL-7 signaling. However, the disadvantage of this new protocol is that the frozen carcasses are not suitable for histological analysis and bone histomorphometry, which require proper fixation, embedding, and sectioning. Suzuki et al. reported a transcriptome analysis of the kidney from mice two

days after their return to Earth from a 31-day stay in the ISS[53]. They found that the spaceflight induced upregulation of uridine diphosphate-glucuronosyltransferase 1A (Ugt1a) isoform genes (Ugt1a1, Ugt1a2, and Ugt1a10) in the kidney. The transcriptome analysis in the present study confirmed upregulation of Ugt1a2 (1.99 fold), but downregulation of Ugt1a1 (0.32 fold) and Ugt1a10 (0.36 fold). The discrepancy between the studies may be due to differences in the duration of microgravity exposure (31 days vs. 9–10 days) and the experimental protocol (live return vs. frozen return).

Because severity of the phosphate-induced kidney damage correlates with phosphate excretion per nephron, mice whose nephron number had been reduced by uninephrectomy suffered from more severe renal tubular damage and interstitial fibrosis than sham-operated mice when the amount of urinary phosphate excretion was the same[1]. In clinical settings, the elderly people and patients with CKD should be more susceptible to the phosphate-induced kidney damage, because the functional nephron number declines with age and CKD progression[54]. CKD is associated with mineral bone disorder (MBD) characterized by bone mineral loss and fracture[55]. In the current understanding of CKD-MBD, CKD is regarded as a cause of bone mineral loss. However, the present study has raised the possibility that bone mineral loss can be a cause of CKD, potentially forming a deterioration spiral toward CKD progression. Indeed, we demonstrated in cross-sectional studies that time spent in sedentary behavior (physical inactivity) correlated independently with low bone mineral density, low estimated glomerular filtration rate (eGFR), and low estimated nephron number in the aged and/or CKD patients[56,57]. We also demonstrated that an exercise therapy by replacing sedentary behavior with physical activity increased bone mineral density[58]. Interventional studies are awaited to determine whether prevention of bone mineral loss by exercise and/or medication may decelerate CKD progression.

In the present study, mice fed a high-phosphate diet containing 1.5% inorganic phosphate for 6 weeks exhibited increased renal expression of several tubular injury markers (Fig. 4), accompanied by histological changes including tubular vacuolization, interstitial fibrosis, and inflammatory cell infiltration (Supplementary Fig. S7). However, these mice did not develop overt renal functional impairment during the observation period. As we have previously reported, a more robust phosphate burden per nephron—such as feeding a 2.0% phosphate diet to uninephrectomized mice for 12 weeks—is required to induce a measurable decline in creatinine clearance. Therefore, the kidney injury induced by dietary phosphate loading in the present study should be regarded as a model of subclinical kidney injury, rather than established CKD. Further studies are warranted to determine whether sustained increases in dietary phosphate intake and/or enhanced bone resorption ultimately lead to overt CKD in clinical settings.

While our findings suggest that bisphosphonates may mitigate kidney injury through inhibition of bone mineral loss, a clinical study has reported an increased risk of acute kidney injury (AKI) associated with bisphosphonate use[59]. A population-based study using a large primary care database found that oral bisphosphonates were associated with a higher incidence of AKI in elderly patients with complex comorbidities. This discrepancy may be partially explained by the known nephrotoxic potential of certain bisphosphonates[60], particularly when administered in patients with advanced age and multiple underlying health conditions. In contrast, clinical trials have demonstrated that bisphosphonates are generally safe in individuals with a creatinine clearance above 30 mL/min[61]. These conflicting observations underscore the need for further prospective studies to evaluate the renal safety and potential therapeutic benefit of bisphosphonates in CKD patients.

## Methods
### Mice
All animal experiments were approved by the Institutional Animal Care and Use Committee of Jichi Medical University, Japan Aerospace Exploration Agency (JAXA), and/or the National Aeronautics and Space Administration (NASA). We have complied with all relevant ethical regulations for

animal use. Mice were housed in a temperature-controlled room (22 ± 2 °C) under a 12-h light/dark cycle with free access to food and water. All animals were maintained under specific pathogen-free (SPF) conditions. Mice were monitored once daily for signs of pain or distress, including reduced mobility, piloerection, and body weight loss. Animals exhibiting more than 20% body weight loss or inability to obtain food or water were considered to have reached humane endpoints. However, no animals reached the pre-defined humane endpoints during the study. All experimental procedures were designed to minimize pain and distress and were conducted in accordance with institutional guidelines for animal care.

## Pharmacological model of bone loss

Mice (C57BL/6J, 8-week-old females) were purchased from Charles River Japan and maintained on a standard chow containing 0.35% inorganic phosphate and tap water. After 2 weeks of acclimatization, mice were administered with recombinant human sRANKL (2.0 mg/kg, Oriental Yeast, Tokyo, Japan) or the same volume of vehicle (phosphate-buffered saline containing 1 mM EDTA) by intraperitoneal injection once (the single dose regimen) or three times for every 24 h (the triple dose regimen). In the single dose regimen, urine was collected every 2 h between 18 and 48 h after the sRANKL injection. Blood and femurs were collected before and 18, 21, 24, 27, 30, 48 h after the sRANKL injection. In the triple dose regimen, blood, femurs, and kidneys were collected before the first sRANKL injection and 1.5, 12, 24, 48 h after the third sRANKL injection. Femurs were fixed in 70% ethanol. The right kidney was snap frozen in liquid nitrogen and stored at −80 °C until used for RNA extraction. The left kidney was fixed in buffered formalin for immunohistochemical analysis as described below. Monosodium risedronate hemipentahydrate [1-hydroxy-2-(3-pyridinyl) ethylidene-1, 1-diphosphonate hemipentahydrate] was purchased from Tokyo Chemical Industry (Tokyo, Japan). Mice were administered with risedronate (10 μg/kg) or the same volume of vehicle (H$_2$O) by subcutaneous injection every 24 h for 5 consecutive days. Mice were transferred individually to metabolic cages to measure food and water consumption and to collect urine for 2–4 days.

## A model of kidney injury induced by dietary phosphate load

Mice (C57BL/6J, 4-week-old females) were maintained on a standard chow diet containing 0.35% inorganic phosphate. After a 2-week acclimatization period, the mice were either switched to a high phosphate diet containing 1.5% inorganic phosphate to induce tubular injury, or continued on the standard chow. At 6, 8, and 10 weeks of age, mice in each group received intraperitoneal injections of either recombinant human sRANKL (2.0 mg/kg) or an equal volume of vehicle (phosphate-buffered saline with 1 mM EDTA), following the triple dose regimen. All mice were euthanized at 12 weeks of age for collection of blood and kidney samples.

## Non-pharmacological model of bone loss (space experiments)

Mice (C57BL/6J, 12-week-old males) were purchased from Jackson Laboratory and acclimated on diet containing 1.3% inorganic phosphate for 4 weeks on the ground. Six (6) mice of similar body weight (28.5 ± 0.5 g, mean ± s.d.) were selected 4 days prior to launch to the International Space Station (ISS) and transferred to the Transportation Cage Unit (TCU)[62], which had syringe-shaped rooms in which mice were individually housed. The TCU was equipped with LED lights for day/night cycles, a temperature/humidity logger, and a syringe-shaped food and a drink nozzle to allow the mice free access to food and water. The mice in the TCU were launched using SpaceX-22 from the NASA Kennedy Space Center (KSC) on June 3rd, 2021 (Day 0) and then transferred to Kibo, the Japanese experimental module attached to the ISS. On Day 9 or Day 10, the mice were starved for 5–16 h, anesthetized by intraperitoneal injection of Ketamine (150 mg/kg) and Xylazine (45 mg/kg), and then sacrificed by whole blood sampling with cardiac puncture and cervical dislocation. The blood samples were transferred to plastic plasma separation tubes containing lithium heparin and centrifuged at $2750 \times g$ for 13 min. After centrifugation, the whole tubes were placed in the −95 °C freezer in the ISS. The carcasses were also stored in the -95°C freezer. The frozen blood samples and carcasses arrived at KSC on July 20th, 2021 via spaceX-22 and then sent to Jichi Medical University via JAXA for analysis. The difference in the food weight between before and after the space flight was measured to calculate the average daily food consumption. As a control group for this microgravity group, mice were housed on ground under the same conditions as in the space (the normal gravity group). Namely, mice (C57BL/6J, 12-week-old males) were acclimatized on diet containing 1.3% inorganic phosphate for 4 weeks at JAXA Tsukuba in Japan. Six (6) mice with the similar body weight as those sent to the ISS (28.5 ± 1.5 g, mean ± s.d.) were selected and transferred individually to the TCU. These mice were fed the same amount of food each day as the average daily food intake of the microgravity group. The blood samples and carcasses were prepared 9 or 10 days later and stored frozen in a −80 °C freezer.

## Blood and urine analyses

All mice were starved at least for 5 h before blood sampling. Plasma FGF23 levels were measured using FGF23 ELISA (Kainos, Tokyo, Japan) according to the manufacturer's protocols. Plasma/urine glucose, phosphate and calcium levels were measured using Fuji Dri-Chem slides and the analyzer (Dri-Chem NX500V, Fuji). Urine creatinine was measured by using DeterminerL CRE (Minaris Medical, Tokyo, Japan) with Varioskan LUX (Thermo Fisher Scientific, MA, USA). Plasma TRACP-5b levels were measured using Mouse TRAP ELISA (Immunodiagnostic, Boldon, UK).

## Measurement of plasma CPP levels

Plasma CPP levels were measured using the gel-filtration method as reported previously[63]. Briefly, heparinized plasma samples were snap-frozen in liquid nitrogen and stored at −80 °C. The stored plasma samples were dissolved at room temperature and then inoculated with bisphosphonate conjugated with an infrared fluorescent dye that binds to calcium-phosphate crystals (OsteoSense 680EX, PerkinElmer). The mixture was incubated at 25 °C for 60 min and then applied to a gel-filtration spin column to separate CPPs from unbound OsteoSense. The amount of CPPs in the plasma sample was expressed as the fluorescent intensity of the flow-through quantified using an infrared fluorescence scanner (Odyssey CLx, LI-COR).

## Bone analysis

For three-dimensional micro computed tomography (μCT) analyses, right femurs were fixed in 70% ethanol for 1 week or over. μCT scanning was performed using a CosmoScan GX II (Rigaku Corporation, Tokyo, Japan; 90 kV, 88 mA) using a voxel size of $10 \times 10 \times 10$ μm. Three-dimensional microstructural image data were reconstructed, and structural indices were calculated using Amira 3D (Thermo Fisher Scientific). After the μCT analysis, the femurs were subjected to the histomorphometric analysis according to a standard procedure[64]. The distal femur was measured in an area (0.6–0.8 mm$^2$) about 500 μm apart from the growth plate using the analysis software WinROOF2013.

## Immunohistochemistry

After removal of renal capsule, kidneys were fixed in buffered 10% formalin solution overnight and then embedded in paraffin wax. Transverse sections of 4 μm thickness were subjected to immunohistochemical staining of renal tubular damage markers including interleukin-36α (IL-36α) and osteopontin protein. Briefly, the sections were deparaffinized in xylene and rehydrated in a graded ethanol series. Endogenous peroxidase activity was blocked with 3% hydrogen peroxide in distilled water. For antigen retrieval, the sections were boiled in Target Retrieval Solution (Dako, Agilent Pathology Solutions, Santa Clara, CA, USA) for 15 min. The sections were blocked with phosphate buffered saline (PBS) containing 1% bovine serum albumin and 10% rabbit serum for IL-36α or a blocking reagent in Histofine Simple Stain Mouse MAX PO (Nichirei Corporation, Tokyo, Japan) for osteopontin. After blocking, the sections were incubated with a goat polyclonal antibody against mouse IL-36α (diluted 1: 100 with PBS containing

1% BSA; AF2297; R&D Systems, Minneapolis, MN, USA), a mouse monoclonal antibody against mouse osteopontin (diluted 1: 100 with PBS containing 1% BSA; sc-21742; Santa Cruz, Dallas, TX, USA), or rat monoclonal antibody against mouse F4/80 (diluted 1: 100 with PBS containing 1% BSA; GTX26640; GeneTex, San Antonio, TX, USA) at room temperature for 1 h and then with Histofine Simple Stain Mouse MAX PO (Nichirei Corporation) at room temperature for 30 min. Immunoreactivity was visualized by incubation with the DAB substrate kit (Dako; Agilent Pathology Solutions). The sections were counterstained with hematoxylin. Images of the stained sections were digitized using a microscope (BX-51; Olympus, Tokyo, Japan).

### Ex vivo imaging of CPP
Ex vivo imaging of CPP in the renal tubular fluid was performed as described previously. Briefly, about 6 h after the third injection of sRANKL or vehicle in the triple dose regimen, the mice were injected with OsteoSense 680EX (Perkin Elmer, 50 nmol/kg) and Hoechst 33342 (20 mg/kg) from tail vein 30 min prior to the imaging to visualize CPP and cell nuclei, respectively. To visualize the interstitial space and the apical membrane of proximal tubules, FITC-labeled Lutus Tetragonolobus Lectin (80 mg/kg) was injected 10 minutes prior to the imaging.

### RNA preparation
The frozen kidneys were homogenized with RNAiso Plus (Takara, Tokyo, Japan). The lysate was transferred to microcentrifuge tube and extracted with one-fifth volume of chloroform. RNA in the aqueous phase was precipitated with an equal volume of isopropanol, washed with 75% ethanol, and dissolved in RNase-free water. For the spaceflight mice, the frozen carcasses were sliced with a thread saw to make serial transverse sections while frozen. Renal tissue was scraped from the surface of the cross section with a carving knife chilled in liquid nitrogen. The scraped renal tissue and left femurs were used for RNA preparation as described above.

### RNA sequencing
The bone RNA samples prepared individually from 6 mice each in the microgravity group and the normal gravity group were mixed in equal amounts per group, and subjected to RNA sequencing analysis. Sequencing libraries were constructed using TruSeq Stranded mRNA Sample Prep Kit (Illumina). High-throughput RNA sequencing were carried out using Illumina NovaSeq 6000 platform. Raw data were aligned to the mouse reference genome GRCm38/mm10 with STAR (v2.7.3a)[65]. Assembly and quantification of the transcripts were accomplished with RESM (v1.3.1)[66]. Transcripts Per Million (TPM) were used for the measurements of the relative abundance of the transcripts. The genes with low expression (TPM lower than 1 in both groups) were eliminated from further analysis. The ratio of the TPM in the microgravity group to the TPM in the normal gravity group was calculated for each transcript. When the ratio was equal to or greater than 1.5, the gene was designated as an upregulated gene by microgravity ($N = 157$). When the ratio was equal to or smaller than 1/1.5, the gene was designated as a downregulated gene by microgravity ($N = 165$). The lists of upregulated and downregulated genes were subjected to enrichment analysis using Metascape (https://metascape.org/gp/index.html#/main/step1).

### Plasma metabolomics
**Sample preparation.** Plasma samples from 6 spaceflight mice and 6 ground control mice were processed for hydrophilic metabolite profiling. Plasma (25 µL) was mixed with 1030 µL of extraction solvent containing internal standards, vortexed, and subjected to ultrasonic extraction under light-protected conditions. The mixture was centrifuged at $16,000 \times g$ for 5 min at 4 °C, and 800 µL of the supernatant was transferred to a new tube. Chloroform ($CHCl_3$, 220 µL) and Milli-Q water (220 µL) were added, vortexed for 1 min, and centrifuged at $16,000 \times g$ for 20 min at 20 °C. The upper (aqueous) layer (400 µL) was collected for analysis of hydrophilic metabolites and dried using a refrigerated centrifugal

concentrator (Labconco, USA) for 3 h. The residue was reconstituted in 50 µL of Milli-Q water and diluted 5-fold with 50% (v/v) acetonitrile to obtain the water-soluble fraction for LC/IC-MS analysis.

**Anionic metabolite profiling (IC–Orbitrap MS; IC-QEMS).** Anions were analyzed on a Thermo Dionex ICS-5000$^+$ ion chromatography system (Thermo Fisher Scientific, Waltham, MA, USA) equipped with an IonPac AS11-HC column (0.4 mm × 250 mm, 4 µm; Thermo Fisher Scientific). Potassium hydroxide (KOH) was used as the eluent at 0.02 mL/min with the following gradient (time, KOH concentration in mM): 0 min (1) – 2 min (1) – 15 min (20) – 32 min (100) – 36 min (100) – 36.1 min (1) – 40.1 min (1). Column temperature was maintained at 35 °C and the suppressor current at 25 mA. Injection volume was 0.4 µL. A sheath liquid (isopropanol containing 0.1% [v/v] acetic acid) was delivered at 5 µL/min to the mass spectrometer. The IC was hyphenated to a Q Exactive Plus Orbitrap mass spectrometer (Thermo Fisher Scientific) operated in negative ESI mode with the following settings: sheath gas 20 (arb), auxiliary gas 10 (arb), spray voltage 4.0 kV, capillary temperature 300 °C, auxiliary gas heater 300 °C, S-lens RF level 50. Full-scan resolution was 70,000 (at m/z 200), mass range m/z 70–1000, AGC target $3 \times 10^6$, and maximum injection time 100 ms.

**Cationic metabolite profiling (HILIC–Orbitrap MS; LC-QEMS).** Cations were analyzed on an Agilent 1290 Infinity LC system using a HILIC-Z column (150 × 2.1 mm, 2.7 µm; Agilent Technologies) held at 40 °C. Mobile phase A was 20 mM ammonium formate with 0.25% formic acid (v/v), and mobile phase B was acetonitrile mixed with 200 mM ammonium formate and 2.5% formic acid at a 9:1 ratio (v/v). The flow rate was 0.25 mL/min. The gradient of %B was: 0 min (100%) → 15 min (70%) → 20 min (10%) → 23 min (10%), followed by a 7-min post-run. Injection volume was 1 µL. The LC was coupled to a Q Exactive Plus Orbitrap mass spectrometer operated in positive ESI. For untargeted/quantitative acquisition we used Full MS with SIM windows: sheath gas 40, auxiliary gas 10, spray voltage 3.5 kV, capillary temperature 250 °C, auxiliary gas heater 300 °C, S-lens RF level 35, resolution 35,000 (at m/z 200), mass range m/z 50–750, AGC target $3 \times 10^6$, and maximum injection time 200 ms. For targeted acquisition we used PRM (parallel reaction monitoring) in positive mode with an inclusion list at m/z 166.0863, 104.0706, 118.0863, 132.1019, and 182.0482; resolution 17,500 (at m/z 200), AGC target $2 \times 10^5$, and maximum injection time 100 ms.

**Quantification of hydrophilic metabolites.** Calibration standards were analyzed at the beginning of each sequence under the corresponding chromatographic method (IC-QEMS or LC-QEMS). Plasma concentrations were calculated by internal-standard calibration according to Eq. (1), setting the internal-standard (IS) concentration in the plasma–extract suspension to 1.321 µM.

$$C_{plasma}(\mu M) = \frac{\frac{A_{sample}}{A_{IS,sample}}}{\frac{A_{STD}}{A_{IS,STD}}} \times C_{STD} \times \frac{C_{IS,susp}}{C_{IS,STD}} \times \frac{V_{extract} + V_{plasma}}{V_{plasma}}$$

where $A_{sample}$ and $A_{STD}$ are analyte peak areas in the sample and standard, $A_{IS,sample}$ and $A_{IS,STD}$ are the corresponding internal-standard peak areas, $C_{STD}$ is the analyte concentration in the calibration standard, $C_{IS,susp} = 1.321$ µM is the IS concentration in the plasma–extract suspension, $C_{IS,STD}$ is the IS concentration in the standard, $V_{plasma} = 25$ µL, and $V_{extract} = 1030$ µL.

### Quantitative RT-PCR (qPCR)
Reverse transcription of RNA (0.5 µg) was carried out using ReverTra Ace qPCR RT Master Mix with gDNA Remover (FSQ-301; Toyobo, Osaka, Japan) according to the manufacturer's protocol. Quantitative RT-PCRs contained 15 ng of cDNA, 410 nM of each primer, and 6 µL of SYBR Green PCR Master Mix (THUNDERBIRD SYBR qPCR Mix, QPS-201; Toyobo)

in a total volume of 12 μL. The PCR reaction (95 °C for 1 min followed by 40 cycles of 95 °C for 10 s, 60 °C for 50 s) was performed on the Roche LC480 system. Relative mRNA levels were calculated by the comparative threshold cycle method using cyclophilin as an internal control. The forward (F) primers and reverse (R) primers used were as follows:

Cyclophilin, F: TGGAGAGCACCAAGACAGACA, R: TGCCGGA GTCGACAATGAT;

Osteopontin, F: TCCAAAGAGAGCCAGGAGAG, R: GGCTTTGG AACTTGCTTGAC;

Ngal, F: GAAATATGCACAGGTATCCTC, R: GTAATTTTGAA GTATTGCTTGTTT;

IL-36α, F: GGGGGAAATCTTCATCACTGA, R: GAAATCTTG AGAGAGTGCCACA;

Kim-1, F: TGCAGAACGCAGCGGTTGTG, R: TCTGCCCCT CAAGGTCTATC;

Cyp24a1, F: TGCCCTATTTAAAGGCCTGTCT, R: CGAGTTGTG AATGGCACACTT;

Cyp27b1, F: GGTCCCTGAGAGGAGCATCA, R: ACCCCATCCT GTCTTGAGAG;

RANK, F: GAGCAGAACTGACTCTATGC, R: CCTGTGTAG CCATCTGTTGA.

## Statistics and reproducibility

For comparisons between two groups, significance was determined using the Student's $t$ test. For comparisons among more than two groups, one-way analysis of variance (ANOVA) with Tukey's multiple-comparison test, or Kruskal–Wallis's test with Dunn's multiple-comparison test were used. $P < 0.05$ was considered statistically significant ($*P < 0.05$, $^\#P < 0.01$; ns not significant, throughout the paper). All statistical analyses were performed using Prism 10 (GraphPad Software). The number of animals used in each experiment was determined based on the results of our previous similar studies. All the experiments besides the space experiment were repeated at least twice with consistent results.

## Reporting summary

Further information on research design is available in the Nature Portfolio Reporting Summary linked to this article.

## Data availability

All data supporting the findings of this study are available within the paper and its Supplementary Information. Numerical source data underlying all graphs in the article can be found in Supplementary Data 1. RNA-seq data are publicly available in the NCBI Gene Expression Omnibus (GEO) under accession number GSE316312.

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

## Acknowledgements

The authors thank Dr. Masafumi Takahashi (Jichi Medical University) for discussion on the space experiments, Mr. Yukinari Ohsaka, Ms. Mayuka Shiino, and Ms. Taeko Yamauchi (Jichi Medical University) for technical assistance, Ms. Kyoko Nakamura (Jichi Medical University) for administrative assistance, Toshiaki Kokubo and Noriko Kajiwara (JAXA visiting veterinarians) for monitoring mouse health. This work was supported in part by the Japan Society for the Promotion of Science KAKENHI Grant Number JP22H00473, Moonshot R&D Program for Agriculture, Forestry and Fisheries (Funding Agency: Bio-oriented Technology Research Advancement Institution), a grant from Japan Aerospace Exploration Agency (JAXA), and Japan Agency for Medical Research and Development (AMED) Grant Number 25gm1510012h0003.

## Author contributions

H.H., Y.M., Y.I., H.M., and H.K. performed the mouse experiments and outcome assessment. T.K. and N.O. carried out imaging of CPP within renal tubules. T.A., R.O., D.K., and D.S. coordinated the spaceflight experiments with NASA. Y.M. and M.K. supervised the project and provided overall guidance. M.K. analyzed the data and wrote the manuscript. All authors reviewed and approved the final version of the manuscript.

## Competing interests

The authors declare the following competing interests: YM, HK, and MK hold a patent on blood CPP assay (JP2016200445A). All other authors declare no competing interests.
