## [Transparent Peer Review file · Communications Biology]

Bone mineral loss damages renal tubules in mice

Corresponding Author: Professor Makoto Kuro-o

Version 0:

Reviewer comments:

Reviewer #1

(Remarks to the Author)

In this work Hayashi and colleagues report that inducing bone resorption in mice with sRANKL injection or by exposition to microgravity increases phosphate, calcium, calciprotein particles (CPP) and FGF23 serum concentrations. Sustained bone resorption induced by sRANKL injections for three days was associated with an increase in the expression of renal tubule damage markers in the absence of serum creatinine concentration modifications. Risedronate treatment partially prevent sRANKL-induced FGF23 increase and renal tubule damages.

Remarks:

1- This study supports the possibility that bone resorption stimulates FGF23 productions by releasing phosphate and calcium. However the authors overstep some of their results:

Figure 1b: the authors state that « (BV/TV) in the femurs showed a decreasing trend up to 24 hours after the sRANKL injection ». This affirmation is not supported by the statistical analysis.

The involvement of CPP in the increase of FGF23 is plausible but it is based on results provided in the literature, not by data presented in the present paper. The authors wrote I quote: « increase in plasma CPP levels (Figure 1e), followed by an increase in plasma FGF23 levels (Figure 1f) ». There is no evidence in this report that CPP increase precedes the rise in FGF23 concentration. It has been reported that CPP may be involved in the stimulation of FGF23 production (Calciprotein particles regulate fibroblast growth factor-23 expression in osteoblasts Ken-ichi Akiyama Kidney International (2020) 97, 702–712.) On the other hand FGF23 can stimulate fetuin production and therefore could be responsible for the elevation of CPP production (FGF23-regulated production of Fetuin-A (AHSG) in osteocytes D. Mattinzoli Bone <http://dx.doi.org/10.1016/j.bone.2015.10.008>).

The following sentence: « The decrease in PTH may be due not only to the increase in plasma calcium levels but also to the increase in plasma FGF23 levels, as FGF23 suppresses PTH expression and secretion » should not be in the results but in the discussion, this assertion is speculative.

2- The mechanisms by which bone resorption might induce renal tubule damage is not clear.

The increases in mRNA of 3 markers associated with renal lesions seems very transient (figure 3). Did the authors perform further analysis of renal tubule damages as measurements of glycosuria or aminoaciduria for example?

Have the authors checked if an increase in phosphate excretion, as observed during phosphate load for instance, could reproduce the renal lesions?

Were Ngal and Kim1 mRNA expressions increased in the mice subjected to microgravity?

The authors hypothesise that the renal damage was due to increased renal phosphate excretion however it is unclear figure 5g if risedronate treatment significantly decreases urinary phosphate excretion. In risedronate treated mice FGF23 concentration was still above control (figure 5f) and seems effective on the kidney since Cyp27b1 and Cyp24a1 expressions were not normalised (figure S7 D, E). Did the author perform renal histology in mice treated with risedronate as shown in figure S4?

As mentioned above the authors cannot mention « an increasing trend of interleukin-36a expression (Figure 4h) in the kidney », since statistical analysis showed no significant difference.

3- Figure 5e: Were Rank mRNA levels significantly different in risedronate treated mice and in control?

Reviewer #2

(Remarks to the Author)

In this manuscript, Hayashi and colleagues explore the connection between bone resorption and kidney injury, focusing on the role of fibroblast growth factor-23 (FGF23). The authors propose that increased bone resorption mobilizes phosphate into the circulation, leading to phosphate overload, CCP formation in renal tubules, and subsequent renal injury. While the study incorporates innovative aspects, particularly the use of microgravity experiments, several critical issues need to be addressed:

Major Concerns:

1. Insufficient Evidence for Renal Injury:

o The current data, which primarily rely on transient changes in renal injury markers, do not convincingly demonstrate meaningful kidney damage. Additional *in vivo* experiments using AKI, AKI-to-CKD, or CKD models with sRANKL and/or bisphosphonate treatment are required to establish a causal link between the proposed mechanism and renal injury.

2. Microgravity Experiment and Potential Confounders:

o While the microgravity experiment is innovative, the use of food intake matching as a control strategy introduces numerous confounders, including:

Altered metabolism and energy expenditure

Exposure to cosmic radiation, which affects inflammation and mineral metabolism

Changes in gastrointestinal nutrient absorption

o Given these factors, how can the authors ensure that dietary phosphate absorption was comparable between spaceflight and control conditions?

3. Role of Kidney Glycolysis and Glycerol-3-Phosphate in Phosphate Sensing:

o Recent studies have identified kidney glycolysis and glycerol-3-phosphate as key regulators of phosphate sensing and FGF23 production (PMID: 32065590, 36821389). Given the metabolic alterations in space, how can the authors exclude the possibility that changes in glycerol-3-phosphate, rather than bone resorption, underlie the observed effects? At a minimum, these studies should be cited and discussed. Especially since μ CT experiments failed to demonstrate significant differences in bone formation.

4. Insufficient Renal Phenotyping in Spaceflight Study:

o The only renal injury assessment in the spaceflight study consists of qPCR analysis. While logistical constraints are acknowledged, this level of analysis is insufficient to support the manuscript's claims. Additional histological or functional kidney assessments would be required for robust conclusions.

5. Discrepancy Between Proposed Mechanism and Clinical Data:

o If the proposed mechanism is correct, bisphosphonate treatment should theoretically protect against AKI or slow CKD progression. However, clinical studies suggest otherwise (PMID: 35579494). How do the authors reconcile this discrepancy? These findings should at least be acknowledged and discussed.

Minor Concerns:

• The μ CT analysis failed to detect significant differences in bone structure. Could alternative methodologies or additional readouts improve resolution and provide further support for the bone resorption hypothesis?

Overall, while the study presents an intriguing hypothesis, additional mechanistic experiments and more comprehensive renal phenotyping are necessary to provide conclusive evidence. Addressing these concerns will significantly strengthen the manuscript's impact and scientific validity.

Version 1:

Reviewer comments:

Reviewer #2

(Remarks to the Author)

The authors have thoughtfully addressed all raised concerns through comprehensive additional *in vitro* and *in vivo* assays, substantially strengthening the manuscript.

However, several minor revisions remain necessary:

Critical Terminology Issue:

The characterization of the high-phosphate diet model as "CKD" is inconsistent with standard clinical definitions. The authors demonstrate tubular injury markers but provide no evidence of reduced renal function (i.e., no measured GFR, creatinine, or estimated glomerular filtration rate). According to KDIGO criteria, CKD requires functional impairment or persistent markers of kidney damage. This model more accurately represents phosphate-induced tubular injury or subclinical kidney injury, reflecting the early subclinical injury phase rather than established CKD. We recommend the authors revise terminology accordingly throughout the manuscript and acknowledge in the Discussion that findings represent tubular injury susceptibility rather than overt CKD progression.

Editorial Note:

Line 341 is missing a punctuation mark following "...that the apparent increase in G3P."

Point-by-point responses to the Reviewers' comments

First of all, we would like to express our sincere gratitude to the Editor and the two reviewers for their constructive and valuable comments. The point-by-point responses to the Reviewers' comments (*italic*) are as follows:

Reviewer #1 (Remarks to the Author):

In this work Hayashi and colleagues report that inducing bone resorption in mice with sRANKL injection or by exposition to microgravity increases phosphate, calcium, calciprotein particles (CPP) and FGF23 serum concentrations. Sustained bone resorption induced by sRANKL injections for three days was associated with an increase in the expression of renal tubule damage markers in the absence of serum creatinine concentration modifications. Risedronate treatment partially prevent sRANKL-induced FGF23 increase and renal tubule damages.

Remarks:

This study supports the possibility that bone resorption stimulates FGF23 productions by releasing phosphate and calcium. However the authors overstep some of their results:

1. Figure 1b: the authors state that « (BV/TV) in the femurs showed a decreasing trend up to 24 hours after the sRANKL injection ». This affirmation is not supported by the statistical analysis.

As the reviewer correctly pointed out, the decrease in BV/TV following the single-dose sRANKL regimen did not reach statistical significance. Accordingly, we intentionally described this observation as “a decreasing trend.” To avoid any misunderstanding and to clearly reflect the statistical outcome, we have revised the text as follows:

Line 104-110 “Concurrently, the trabecular bone volume over the tissue volume (BV/TV) in the femurs showed a decreasing trend up to 24 hours after the sRANKL injection, although this change did not reach statistical significance (Figure 1b). As the bone mineral density was not changed (Figure 1c), we hypothesized that a statistically significant loss of bone mineral could be achieved by intensifying the sRANKL treatment. Nevertheless, plasma levels of phosphate (Figure 1d) and calcium (Supplementary Figure S1A) showed a transient but statistically significant elevation.”

2. The involvement of CPP in the increase of FGF23 is plausible but it is based on results

provided in the literature, not by data presented in the present paper. The authors wrote I quote: « increase in plasma CPP levels (Figure 1e), followed by an increase in plasma FGF23 levels (Figure 1f) ». There is no evidence in this report that CPP increase precedes the rise in FGF23 concentration.

As the reviewer noted, the proposed involvement of CPP in the regulation of FGF23 is based on previously published data; however, we would like to respectfully clarify that the referenced study was conducted by our own laboratory (Akiyama K, Miura Y, Hayashi H, Sakata A, Matsumura Y, Kojima M, Tsuchiya K, Nitta K, Shiizaki K, Kurosu H, Kuro-o M, Calciprotein particles regulate fibroblast growth factor-23 expression in osteoblasts. *Kidney Int.* 2019, 97, 702-712). Therefore, although the data reported in this paper are not presented again in the current manuscript, they do originate from our laboratory and directly support the conceptual framework of the present study.

In our time-course analysis, plasma CPP levels showed a statistically significant increase from baseline as early as 18 hours after sRANKL injection, but then began to decline and were no longer significantly different from baseline at 24 hours and beyond (Figure 1e). In contrast, plasma FGF23 levels first showed a statistically significant increase at 21 hours and remained elevated until 30 hours after the sRANKL injection (Figure 1f). These data indicate that the significant rise in CPP levels preceded the significant increase in FGF23 and that the FGF23 levels remained elevated even after the CPP levels had declined. If we had not published the paper showing that CPP directly induced FGF23 expression in osteoblasts, we agree that these data alone do not establish a causal relationship. However, to clarify that this is a temporal association, we revised the text as follows:

Line 111-113“, which was associated with a transient increase in plasma CPP levels at 18 hours (Figure 1e). Plasma FGF23 levels first showed a statistically significant increase at 21 hours and remained elevated until 30 hours after the sRANKL injection (Figure 1f).

Furthermore, the following sentences were added to Discussion:

Line 299-303 " After promotion of bone resorption by sRANKL administration, the rise in plasma CPP levels preceded the significant increase in FGF23 and that the FGF23 levels remained elevated even after the CPP levels had declined (Figures 1e, 1f). This temporal association was consistent with the finding that CPP induced FGF23 expression, which we have reported previously and therefore did not include the original experimental data supporting this finding in the present study."

3. *It has been reported that CPP may be involved in the stimulation of FGF23 production (Calciprotein particles regulate fibroblast growth factor-23 expression in osteoblasts Ken-ichi Akiyama Kidney International (2020) 97, 702–712.) On the other hand FGF23 can stimulate fetuin production and therefore could be responsible for the elevation of CPP production (FGF23-regulated production of Fetuin-A (AHSG) in osteocytes D. Mattinzoli Bone <http://dx.doi.org/10.1016/j.bone.2015.10.008>).*

We reported in our previous publication that CPP stimulated FGF23 production (Akiyama K, et al. Calciprotein particles regulate fibroblast growth factor-23 expression in osteoblasts. *Kidney Int.* 2019, 97, 702-712). Conversely, FGF23 has also been reported to induce the production of fetuin-A, which in turn could promote CPP formation (Mattinzoli D, et al. FGF23-regulated production of Fetuin-A (AHSG) in osteocytes. *Bone* 2016, 83, 25-47). We acknowledge that the interaction between CPP and FGF23 may be bidirectional. However, as described above, our data (Figures 1e and 1f) show that the increase in plasma CPP levels preceded the significant elevation in FGF23, and that FGF23 levels remained elevated even after CPP levels had returned to baseline. Based on this temporal relationship, we consider it unlikely that the observed increase in FGF23, even if it had upregulated fetuin-A expression, was responsible for the transient rise in CPP levels following sRANKL injection.

4. *The following sentence: « The decrease in PTH may be due not only to the increase in plasma calcium levels but also to the increase in plasma FGF23 levels, as FGF23 suppresses PTH expression and secretion » should not be in the results but in the discussion, this assertion is speculative.*

These sentences were moved to Discussion (Line 303-306).

The mechanisms by which bone resorption might induce renal tubule damage is not clear.

5. *The increases in mRNA of 3 markers associated with renal lesions seems very transient (figure 3). Did the authors perform further analysis of renal tubule damages as measurements of glycosuria or aminoaciduria for example?*

We measured urinary glucose excretion and found no significant difference between sRANKL-treated and vehicle-treated mice. The data are presented in the newly added Supplementary Figure S4, and the following sentence has been incorporated into the main text:

Line 155-157 “However, the renal tubular damage was not associated with an increase in plasma creatinine levels (Figure 3f) or urinary glucose excretion (Supplementary Figure S4), suggesting that the transient tubular injury induced by sRANKL treatment did not lead to significant impairment of tubular function.”

6. Have the authors checked if an increase in phosphate excretion, as observed during phosphate load for instance, could reproduce the renal lesions?

We have described in the Introduction the mechanism by which phosphate loading induces kidney damage, as established in our previous publications (Shiizaki K, Tsubouchi A, Miura Y, Seo K, Kuchimaru T, Hayashi H, Iwazu Y, Miura M, Battulga B, Ohno N, Hara T, Kunishige R, Masutani M, Negishi K, Kario K, Kotani K, Yamada T, Nagata D, Komuro I, Itoh H, Kurosu H, Murata M, Kuro-o M, Calcium phosphate microcrystals in the renal tubular fluid accelerate chronic kidney disease progression. *J Clin Invest.* 2021, 131, e145693). Furthermore, we conducted additional experiments using CKD mouse models induced by dietary phosphate loading. Consistent with our earlier publications, mice fed a high-phosphate diet (1.5% inorganic phosphate) exhibited elevated expression of tubular injury markers compared with those fed a normal-phosphate diet (0.35% inorganic phosphate). These data were shown in the newly added Figure 4.

7. Were Ngal and Kim1 mRNA expressions increased in the mice subjected to microgravity?

Although osteopontin expression was increased (Figure 5g in the revised figures), Ngal and Kim1 mRNA levels were not significantly increased under microgravity.

8. The authors hypothesise that the renal damage was due to increased renal phosphate excretion however it is unclear figure 5g if risedronate treatment significantly decreases urinary phosphate excretion.

We respectfully disagree with the reviewer’s comment that “it is unclear in Figure 5g whether risedronate treatment significantly decreases urinary phosphate excretion.” As shown in Figure 6g (Figures were re-numbered because a new figure was added to the revised manuscript), a clear and statistically significant reduction in urinary phosphate excretion is observed in risedronate-treated mice compared to sRANKL-only mice, both on day 0 and day 1. This is evident when comparing the dark gray bars (sRANKL alone) and the light gray bars (sRANKL + risedronate) in the figure.

9. In risedronate treated mice FGF23 concentration was still above control (figure 5f) and seems effective on the kidney since Cyp27b1 and Cyp24a1 expressions were not normalised (figure S7 D, E).

As the reviewer pointed out, risedronate treatment significantly attenuated, but did not completely normalize, the sRANKL-induced elevation of plasma FGF23 levels (Figure 6f). Consequently, residual FGF23 activity may still have influenced renal vitamin D metabolism, which could explain why Cyp27b1 and Cyp24a1 expression levels were not normalized (Figure S10D, E). This is consistent with the observation that the inhibitory effect of risedronate on sRANKL-induced osteoclast activation was only partial, as indicated by the attenuation—but not full restoration—of TRACP-5b levels (Figure 6b).

10. Did the author perform renal histology in mice treated with risedronate as shown in figure S4?

According to the reviewer's suggestion, we performed renal histology using hematoxylin-eosin (H&E) staining. Mice treated with sRANKL showed prominent interstitial inflammatory cell infiltration, whereas this change was not observed in mice co-treated with sRANKL and risedronate. Consistent with the histology, risedronate significantly reduced renal IL-6 mRNA expression compared with sRANKL alone. We have added representative H&E images and qPCR data for IL-6 expression to Figure 7f and Figure 7e, respectively, and the following sentence has been incorporated into the main text:

Line 243-248 “These changes induced by the risedronate treatment were associated with significant decrease in expression of markers for renal tubular damage and inflammation at the mRNA and protein levels (Figure 7). Consistent with this, the risedronate treatment alleviated sRANKL-induced interstitial cell infiltration (Figure 7f), whereas it did not affect either plasma levels of phosphate, calcium, and creatinine or renal expression levels of the Cyp24a1 and Cyp27b1 genes in mice treated with sRANKL (Supplementary Figure S10).”

11. As mentioned above the authors cannot mention « an increasing trend of interleukin-36a expression (Figure 4h) in the kidney », since statistical analysis showed no significant difference.

As the reviewer correctly pointed out, the increase in IL-36 α expression in the kidney under microgravity did not reach statistical significance (P = 0.178 as shown in Figure 5h).

Accordingly, we intentionally described this observation as “an increasing trend.” To avoid any misunderstanding and to clearly reflect the statistical outcome, we have revised the text as follows:

Line 230-232 “An increasing trend in expression levels of another tubular damage marker interleukin-36 α was observed, although this change did not reach statistical significance (Figure 5h). “

12. Figure 5e: Were Rank mRNA levels significantly different in risedronate treated mice and in control?

No, they were not. To clearly reflect this, pairwise comparison lines were added to Figure 5e.

Reviewer #2 (Remarks to the Author):

In this manuscript, Hayashi and colleagues explore the connection between bone resorption and kidney injury, focusing on the role of fibroblast growth factor-23 (FGF23). The authors propose that increased bone resorption mobilizes phosphate into the circulation, leading to phosphate overload, CCP formation in renal tubules, and subsequent renal injury. While the study incorporates innovative aspects, particularly the use of microgravity experiments, several critical issues need to be addressed:

Major Concerns:

1. Insufficient Evidence for Renal Injury:

The current data, which primarily rely on transient changes in renal injury markers, do not convincingly demonstrate meaningful kidney damage. Additional in vivo experiments using AKI, AKI-to-CKD, or CKD models with sRANKL and/or bisphosphonate treatment are required to establish a causal link between the proposed mechanism and renal injury.

We appreciate the reviewer’s comment. Mice treated with sRANKL showed a transient upregulation of renal tubular injury markers in the absence of overt functional impairment, a pattern that parallels subclinical acute kidney injury (AKI) in humans. This model is clinically relevant, as subclinical AKI is recognized as a risk factor for the progression to CKD.

According to the reviewer's suggestion, we performed additional experiments to determine whether increased bone resorption induced by sRANKL may cause meaningful renal injury in a mouse CKD model.

As we previously reported, mice fed a high-phosphate diet develop sustained renal injury characterized by tubular damage and interstitial inflammation and fibrosis over the course of several weeks, serving as a model of CKD (Shiizaki K, et al. Calcium phosphate microcrystals in the renal tubular fluid accelerate chronic kidney disease progression. *J Clin Invest* 131, e145693, 2021). In the present study, we placed mice on a high phosphate diet containing 1.5% inorganic phosphate, starting at 6 weeks of age and continuing for 6 weeks (the newly added Figure 4). Under this experimental condition, the expression levels of tubular injury markers (osteopontin, IL-36 α , Kim-1, and Ngal) were significantly increased (Figure 4b, c and Supplementary Figure S6A, B). In contrast, the expression levels of inflammation markers (F4/80, Mcp-1, TNF α , and IL-6) tended to increase but did not reach statistical significance (Figure 4d, e and Supplementary Figure S6C, D). Using this mild CKD model, we administered sRANKL in the triple-dose regimen every other week to assess whether CKD mice would develop more sustained and severe kidney injury compared to control mice by the increased bone resorption. In control mice, the expression levels of tubular injury markers returned to baseline within 10 days after the final sRANKL injection. In contrast, the expression levels of several markers for tubular injury and inflammation were further increased in CKD mice at the same time point. Two-way ANOVA revealed significant interactions between CKD and sRANKL treatment for osteopontin, IL-36 α , and Mcp-1, indicating that endogenous phosphate mobilized by sRANKL and exogenous phosphate supplied by diet act synergistically to exacerbate tubular injury and interstitial inflammation.

We showed the data obtained by these additional experiments as new figures (Figure 4 and Supplementary Figure S6), summarized the discussion above, and added to the revised text (Line 169-194, 274-286, 468-475, 979-986).

2. Microgravity Experiment and Potential Confounders:

While the microgravity experiment is innovative, the use of food intake matching as a control strategy introduces numerous confounders, including:

- *Altered metabolism and energy expenditure*
- *Exposure to cosmic radiation, which affects inflammation and mineral metabolism*
- *Changes in gastrointestinal nutrient absorption*

Given these factors, how can the authors ensure that dietary phosphate absorption was comparable between spaceflight and control conditions?

We agree with the reviewer that spaceflight and ground conditions differ not only in gravity but also in factors such as cosmic radiation, body fluid distribution, energy expenditure, and gastrointestinal physiology. These factors could, in principle, influence mineral metabolism and nutrient absorption. We and NASA are fully aware that the inability to eliminate such cofounders represents a major scientific limitation of space experiments. In addition to these scientific constraints, there are unavoidable practical limitations, including the restricted number of mice that can be housed aboard the ISS, the limited time astronauts—who are responsible for multiple mission tasks—can devote to animal experiments, and the fact that astronauts are not necessarily proficient in experimental techniques involving mice. While space experiments inevitably entail these limitations, we consider that the data obtained in the present space experiment are consistent with our hypothesis that phosphate released from bone contributes to the induction of renal tubular injury.

In our study, spaceflight mice exhibited significantly higher circulating phosphate, calcium, CPP, and FGF23 levels than ground controls, indicating greater net mineral entry into the circulation. In addition, spaceflight mice showed higher plasma TRAcP-5b and a transcriptomic signature consistent with increased osteoclast activity and reduced bone formation, indicating increased release of endogenous phosphate and calcium from bones. Of note, prior study showed that simulated galactic cosmic rays alone promoted osteoclastogenesis and bone loss under normal gravity. Thus, under space conditions, both microgravity and cosmic radiation contribute to enhanced bone resorption. However, as the reviewer pointed out, it is possible to argue that, for reasons not yet fully understood, intestinal mineral absorption is markedly increased in spaceflight mice and contributes more to the increase in mineral inflow into circulation than does enhanced bone resorption. To evaluate this possibility, we performed plasma metabolomic profiling.

Spaceflight mice exhibited lower plasma levels of citrulline—a recognized biomarker of intestinal enterocyte mass and absorptive function—indicating a likely reduction, rather than enhancement, of nutrient absorption capacity. Moreover, microbiota–host co-metabolites such as p-hydroxybenzoate and phenaceturate were also significantly decreased, a pattern consistent with diminished intestinal uptake (the newly added Supplementary Figure S11A). Collectively, these findings do not support the hypothesis that spaceflight mice experience a generalized enhancement of intestinal nutrient absorption.

Regarding mineral absorption, calcium isotope and tracer studies in astronauts have shown that spaceflight reduces intestinal calcium absorption. If the same holds true for mice, the observation that spaceflight mice nevertheless exhibited higher blood calcium levels suggests that bone-derived calcium influx exceeded any reduction in intestinal uptake. Because bone resorption releases both calcium and phosphate into the circulation, it is reasonable to infer that a substantial amount of phosphate was likewise mobilized from bone in spaceflight mice.

Taken together, and under conditions of matched dietary intake, our findings are consistent with the interpretation that the elevations in blood phosphate and calcium observed in spaceflight mice may be attributable primarily to increased bone resorption rather than to enhanced intestinal absorption. However, we were unable to directly assess intestinal phosphate absorption due to the logistical constraints of spaceflight experiments. In particular, urine collection is not feasible under microgravity. We have acknowledged this as a limitation of the present study and suggest that future work employ isotope tracers and fecal balance analyses to more directly evaluate phosphate absorption under spaceflight versus ground conditions.

The discussion above was added to the revised text (Line 357-394).

3. Role of Kidney Glycolysis and Glycerol-3-Phosphate in Phosphate Sensing:

Recent studies have identified kidney glycolysis and glycerol-3-phosphate as key regulators of phosphate sensing and FGF23 production (PMID: 32065590, 36821389). Given the metabolic alterations in space, how can the authors exclude the possibility that changes in glycerol-3-phosphate, rather than bone resorption, underlie the observed effects? At a minimum, these studies should be cited and discussed. Especially since μ CT experiments failed to demonstrate significant differences in bone formation.

We appreciate the reviewer for raising this important point. To address this concern, we performed plasma metabolome analysis and observed higher glycerol-3-phosphate (G3P) levels in spaceflight mice than in ground controls (Supplementary Figure S11A). However, hemolysis was present in 3 of 6 spaceflight plasma samples. Whole blood sampling by cardiac puncture of anesthetized mice in microgravity is technically demanding for astronauts and likely contributed to hemolysis. Hemolysis releases G3P and glycerophosphorylcholine (GPC) from erythrocytes and promotes enzymatic hydrolysis of GPC to G3P and choline. Consistent with this, the hemolyzed samples showed higher G3P, choline, and GPC than non-hemolyzed samples (Supplementary Figure S11B). These observations suggest that the apparent increase in G3P in

spaceflight mice may, at least in part, reflect a hemolysis-related artifact. Accordingly, whether elevated G3P contributes to increased FGF23 in spaceflight mice remains to be determined.

The discussion above was added to the revised text (Line 341-351).

4. Insufficient Renal Phenotyping in Spaceflight Study:

The only renal injury assessment in the spaceflight study consists of qPCR analysis. While logistical constraints are acknowledged, this level of analysis is insufficient to support the manuscript's claims. Additional histological or functional kidney assessments would be required for robust conclusions.

Unlike prior spaceflight studies in which mice were returned to Earth alive and then dissected, in the present mission the mice were euthanized aboard the ISS, frozen whole, and transported to Earth. This approach was chosen to minimize postflight re-adaptation that could reverse microgravity-induced changes during the several-day interval between landing and dissection. Prioritizing sample integrity, however, precluded conventional histology: thawing whole carcasses to isolate kidneys for histological analysis would introduce freeze-thaw artifacts and tissue degeneration. Instead, we performed plasma metabolomics and observed no differences in creatinine or urea between spaceflight and ground controls, indicating that—within the sensitivity of these markers—renal function was not detectably impaired by spaceflight (Supplementary Figure S11A).

The discussion above was added to the revised text (Line 351-354).

5. Discrepancy Between Proposed Mechanism and Clinical Data:

If the proposed mechanism is correct, bisphosphonate treatment should theoretically protect against AKI or slow CKD progression. However, clinical studies suggest otherwise (PMID: 35579494). How do the authors reconcile this discrepancy? These findings should at least be acknowledged and discussed.

We thank the reviewer for bringing this important study to our attention. The cited study, based on a large primary care database, reported that oral bisphosphonate use was associated with an increased risk of AKI in an elderly population with complex comorbidities. This observation may be attributable, at least in part, to the known nephrotoxicity of certain bisphosphonates. In contrast, clinical trials have demonstrated that bisphosphonates generally exhibit a favorable safety profile in patients with a creatinine clearance above 30 mL/min (Suresh E, et al, Safety

issues with bisphosphonate therapy for osteoporosis. *Rheumatology* 2014, 53, 19-31). These seemingly contradictory findings highlight the need for further well-designed clinical studies to clarify the risk–benefit balance of bisphosphonate therapy in patients with CKD, particularly in those with varying stages of renal function and comorbid conditions.

The following paragraph was added to Discussion (Line 431-440): "While our findings suggest that bisphosphonates may mitigate kidney injury through inhibition of bone mineral loss, a clinical study has reported an increased risk of acute kidney injury (AKI) associated with bisphosphonate use. A population-based study using a large primary care database found that oral bisphosphonates were associated with a higher incidence of AKI in elderly patients with complex comorbidities. This discrepancy may be partially explained by the known nephrotoxic potential of certain bisphosphonates, particularly when administered in patients with advanced age and multiple underlying health conditions. In contrast, clinical trials have demonstrated that bisphosphonates are generally safe in individuals with a creatinine clearance above 30 mL/min. These conflicting observations underscore the need for further prospective studies to evaluate the renal safety and potential therapeutic benefit of bisphosphonates in CKD patients."

Minor Concerns:

6. *The μ CT analysis failed to detect significant differences in bone structure. Could alternative methodologies or additional readouts improve resolution and provide further support for the bone resorption hypothesis?*

We appreciate the reviewer's comment. In previous space experiments, mice were returned alive from the ISS. However, transportation to the laboratory for dissection typically required a few days, raising concerns that the changes induced in microgravity might be reversed by subsequent exposure to normal gravity. Although bone loss was consistently observed in those prior missions, it remains possible that alterations in gene expression and metabolic profiles were modified during the few days of re-exposure to Earth's gravity. To eliminate this possibility, we euthanized the spaceflight mice aboard the ISS and immediately preserved their carcasses by whole-body freezing. After transport to Earth, the carcasses were thawed to fix the bones and prepare histological sections. From a preliminary ground-based experiment, we had recognized that the freeze–thaw process causes degradation of cellular components, thereby precluding reliable bone histomorphometric analysis. As a viable alternative approach, we employed transcriptome profiling by extracting RNA from bone tissue prior to thawing, which revealed gene expression signatures indicative of enhanced bone resorption and suppressed bone formation in spaceflight mice. Thus, although conventional histomorphometry was not

feasible, the transcriptomic data provided an alternative line of evidence supporting our bone resorption hypothesis (Supplementary Figures S8 and S9).

Overall, while the study presents an intriguing hypothesis, additional mechanistic experiments and more comprehensive renal phenotyping are necessary to provide conclusive evidence. Addressing these concerns will significantly strengthen the manuscript's impact and scientific validity.

We appreciate the reviewer's encouraging comments.

Responses to REVIEWERS' COMMENTS:

Reviewer #2 (Remarks to the Author):

The authors have thoughtfully addressed all raised concerns through comprehensive additional in vitro and in vivo assays, substantially strengthening the manuscript.

However, several minor revisions remain necessary:

Critical Terminology Issue:

The characterization of the high-phosphate diet model as “CKD” is inconsistent with standard clinical definitions. The authors demonstrate tubular injury markers but provide no evidence of reduced renal function (i.e., no measured GFR, creatinine, or estimated glomerular filtration rate). According to KDIGO criteria, CKD requires functional impairment or persistent markers of kidney damage. This model more accurately represents phosphate-induced tubular injury or subclinical kidney injury, reflecting the early subclinical injury phase rather than established CKD. We recommend the authors revise terminology accordingly throughout the manuscript and acknowledge in the Discussion that findings represent tubular injury 2 of 3 susceptibility rather than overt CKD progression.

In accordance with the reviewer’s recommendation, we revised the text and Figures (Figure 4 and Supplementary Figure S7) and avoided the use of the term “CKD” throughout the manuscript. In addition, we added the following paragraph to the Discussion to clarify the nature and limitations of the experimental model (Line 438-450).

In the present study, mice fed a high-phosphate diet containing 1.5% inorganic phosphate for 6 weeks exhibited increased renal expression of several tubular injury markers (Figure 4), accompanied by histological changes including tubular vacuolization, interstitial fibrosis, and inflammatory cell infiltration (Supplementary Figure S7). However, these mice did not develop overt renal functional impairment during the observation period. As we have previously reported, a more robust phosphate burden per nephron—such as feeding a 2.0% phosphate diet to uninephrectomized mice for 12 weeks—is required to induce a measurable decline in creatinine clearance. Therefore, the kidney injury induced by dietary phosphate loading in the present study should be regarded as a model of subclinical kidney injury, rather than established CKD. Further studies are warranted to determine whether sustained increases in dietary phosphate intake and/or enhanced bone resorption ultimately lead to overt CKD in clinical settings.